
# The role of catchment characteristics, sewer network, SWMM model parameters in urban catchment management based on stormwater flooding: modelling, sensitivity analysis, risk assessment

Bartosz Szeląg[1], Adam Kiczko[2], Grzegorz Wałek[3], Ewa Wojciechowska[4], Michał Stachura[5], Francesco Fatone[6]

[1] Faculty of Environmental, Geomatic and Energy Engineering, Kielce University of Technology, 25-314 Kielce, Poland
[2] Institute of Environmental Engineering, Warsaw University of Life Sciences-SGGW, 02-797 Warsaw, Poland
[3] Institute of Geography and Environmental Sciences, Jan Kochanowski University in Kielce, 25 – 406, Kielce, Poland
[4] Faculty of Civil and Environmental Engineering, Gdansk University of Technology, 80-233, Gdansk, Poland
[5] Faculty of Law and Social Sciences, Jan Kochanowski University, 25 – 406, Kielce, Poland
[6] Department of Science and Engineering of Materials, Environment and Urban Planning-SIMAU, Polytechnic University of Marche Ancona, 60121 Ancona, Italy

*Correspondence to*: Bartosz Szeląg (bszelag@tu.kielce.pl)

**Abstract.** Sustainable development of urban areas creates an increasing demand for computation tools supporting urban management strategies to mitigate the harmful impact of climate changes on the environment and quality of life, which would at the same time adapt coherently to the latest trends in architecture and urban planning. To date, hydrodynamic models of catchments have been used for this purpose. However, their application is limited due to the costs of model construction and problems with data acquisition. In this study an innovative algorithm for modelling specific flood volume is proposed, which can be applied to assess the need for stormwater network modernisation as well as for advanced flood risk assessment. In contrast to the currently used models, the approach adopted in this study includes characteristics of a catchment and of stormwater network, as well as calibrated model parameters expressing catchment retention and the conductivity of the stormwater network. In the proposed computation method, extended sensitivity analysis was conducted. Sensitivity coefficients of calibrated SWMM (Storm Water Management Model) model parameters were determined with regard to rainfall intensity, catchment and stormwater network characteristics. This extended sensitivity analysis enables an evaluation of the spatial variability of specific flood volume and sensitivity coefficients within a catchment, which is extremely important for identifying the most vulnerable areas threatened by flooding. This allows modernisation work to be focused on areas specifically susceptible to stormwater network failure. The measurement results for a catchment area in Kielce, Poland were used for the presentation of subsequent computation stages of the developed algorithm.

The presented computation method facilitates the management of urban catchments and water resources in urban areas, which can be applied at the stage of urban development planning and be used to inform decisions regarding modernisation and operation of stormwater networks. One of such examples is the demonstration of the reduction of the probability of system failure with the threshold of permissible roughness of sewage pipes. The adopted approach also helps to identify the key characteristics of catchment, stormwater network and SWMM model parameters, which have the highest



impact on the functioning of a stormwater network, which is extremely relevant in terms of assessing and mitigating uncertainty in simulation results.

## 1 Introduction

Climate change and urbanisation are major drivers of increased frequency and severity of hydraulic overloads in urban catchments, leading to flooding events, which cause decrease of life standard, material losses, traffic difficulties etc. (Petit-Boix et al., 2017; Chang et al., 2021). One of the main factors leading to hydraulic overloads is associated with the wearing of storm sewers resulting in increased roughness. Therefore, criteria for assessing stormwater network operating were introduced, which should be taken into consideration both at the design stage and while planning modernisation. According to these

criteria, one of the key assessment parameters is frequency of stormwater flooding (DWA – A118E, 2006; EN – 752, 2006). However, since this parameter has a typically qualitative character, some further modifications were proposed. Based on computation results for stormwater networks, Siekmann and Pinekamp (2011) defined the boundary values of specific flood volume. Exceeding these values should be considered as a clear signal for decision-makers to implement the process of improving stormwater management in the catchment.

Mathematical modelling of the stormwater network provides significant support in a decision-making process. According to (Kirshen et al., 2015; Chen et al., 2016; Mignot et al., 2019), hydrodynamic catchment models are usually applied. For many years, the United States Environment Protection Agency (EPA) has been developing computation tools to simulate stormwater network operation, which allow for implementation of Green Infrastructure units in the catchment. One of the most common tools is the SWMM (Storm Water Management Model) program, which has been used in multiple studies (Baek et

al., 2020; Behrouz et al., 2020). SWMM can be applied for a simplified simulation of stormwater runoff from a catchment, flow rates in stormwater network as well as simulation of hydraulic overloads resulting in flooding (Teng et al., 2017; Cheng et al., 2019). However, in order to establish the range of flooding, stormwater flow directions and water depth, the SWMM source code needs either modification or integration with other calculation tools (Sañudo et al., 2020; Shojaeizadeh et al., 2021). Moreover, the hydrodynamic model of a catchment requires calibration before it can be applied for stormwater

management in a certain catchment area. Data on the spatial development of a catchment, stormwater network and high-resolution rainfall statistics and flow rate measurements are necessary to accomplish the calibration process. Yet, due to the multiplicity of the parameters, the problems encountered at the identification stage are quite frequent, which leads to uncertainty of results (Her et al., 2015; Knighton et al., 2016, Kiczko et al., 2018; Fong and Chui, 2020).

        Optimization methods are frequently used to estimate parameters in hydrodynamic models (De Paola et al., 2016;

Swathi et al., 2018; Awol et al., 2018). Model calibration can be divided into two stages: (1) sensitivity analysis, aimed at eliminating parameters that have an insignificant impact on results, and (2) uncertainty, which analyses interactions between parameters and identifies them based on empirical distribution. According to literature data (Zhang et al., 2019; Cristiano et al., 2019) local and global methods are applied for sensitivity analysis. However, the impact of rainfall intensity is neglected at the calculation stage, which affects model sensitivity results (Razavi and Gupta, 2015; Fatone et al., 2021) and can hinder





an adequate selection of parameters at the calibration stage. Sensitivity analysis is limited to sub-catchments, and so it is impossible to predict the impact of catchment characteristics on calculation results. Moreover, the impact of calibrated parameters of hydrodynamic models in relation to catchment characteristics is unknown. This issue is of major significance for catchment modelling and selecting proper methods for identifying catchment characteristics including retention of impervious and permeable areas, roughness and slope of terrain, as well as for measuring Manning roughness coefficient for

channels (Fraga et al., 2016; Kelleher et al., 2017). The accuracy of these measurements affects the uncertainty of the results obtained and means that some parameters may be discounted. The next stage of model calibration is uncertainty analysis (Chen et al., 2018; Teweldebrhan et al., 2020; Kim et al., 2021). According to Dotto et al. (2014) and Chen et al. (2018) the GLUE method (Generalized Likelihood Uncertainty Estimation) is currently the most frequently used. Computations confirmed that the uncertainty of calibrated parameters of hydrodynamic models has considerable impact on simulation results (Meresa and

Romanowicz. 2017; Kiczko et al., 2018; Szeląg et al., 2022).

Considering the problems associated with calibrating hydrodynamic models, machine learning methods can offer a solution for modelling and assessing how stormwater networks operate. The structure of these models, based on the gathered measurement data, is identified at the so-called learning stage, and the empirical coefficients are also determined. Then, the appointed model is tested using independent data. The calibration of such a model is simpler in comparison to hydrodynamic

models due to the fact that a number of advanced statistical methods are already implemented in computing packages. So, a basic knowledge on their operating is enough to establish a simulation model (Hutchins et al., 2016). A number of machine learning algorithms (boosted tree, random forest, neural network, machine automata) have already been applied in modelling hydraulic overload and flooding in urban catchments, as discussed in detail by (Yu et al., 2015; Ke et al., 2020; Yang et al., 2020). However, until now there have been no attempts to construct models (simulators) that might identify specific flood

volume as a tool for evaluating the efficacy of existing stormwater systems and identifying a need for modernisation. This issue is of tremendous significance because most applied models are defined for single catchments, which means that they are not universal in character and do not offer the possibility of correcting the catchment retention, catchment characteristics or stormwater network. This considerably limits the application of such tools in stormwater management in the catchment.

In our study, a novel algorithm for creating a simulator to predict specific flood volume is developed. In the proposed

approach, the stormwater flooding is related to rainfall data, catchment characteristics and calibrated SWMM model parameters, which enables this tool to be implemented for different catchments with various characteristics, both at the stage of spatial planning and during the daily operation of stormwater networks. In the adopted algorithm, an innovative sensitivity coefficient was defined, enabling the impact of rainfall intensity as well as catchment and stormwater network characteristics on the computation results of sensitivity coefficients to be evaluated with regard to calibrated SWMM model parameters.

Subsequent computation stages of the algorithm based on the measurement data from an urban catchment in Kielce are presented in the article.

**2 Case study**

The analysed urban catchment is located in the south-eastern part of Kielce, central Poland, Świętokrzyskie region

(Fig.1). Residential districts, public buildings, main and side streets are located in the study area. The catchment area covers

63 ha and consists of 40% impervious and 60% permeable areas. The road density is 108 m·ha⁻¹ (Wałek, 2019), and the terrain

denivelation is 11.20m (the ordinates of the highest and the lowest points of the terrain are 271.20 m and 260 m above sea

level, respectively). The length of the main interceptor channel in the stormwater network is 1569 m, with an average slope of

0.71%. The diameter of the main interceptor expands from 600 to 1250 mm, while the diameters of side sewers vary between

300 and 1000 mm. The slope of the sewers varies between 0.04 and 3.90%.

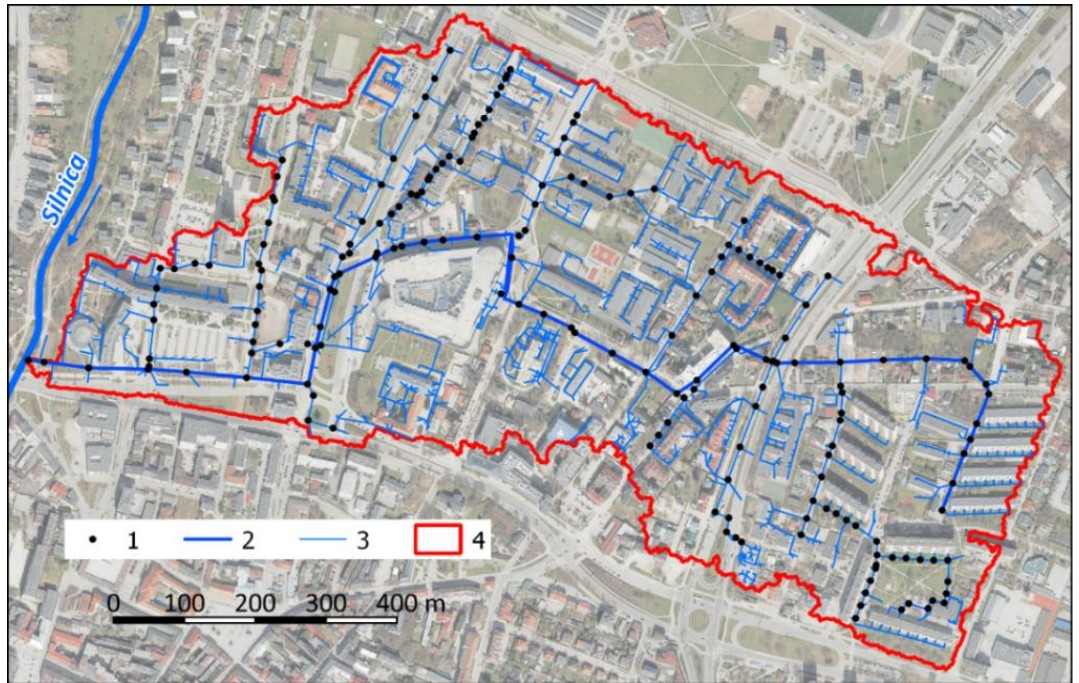

**Figure. 1. Study catchment area (Wałek, 2019).**

The analysed stormwater system is separated from the municipal sewage. Stormwater flows to the division chamber (DC), and

after reaching a depth of 0.42 m it flows into a stormwater treatment plant (STP). The treated stormwater outflows into the

Silnica river. During heavy rainfall, when the stormwater level in the DC exceeds the overflow level (OV), it is discharged by

the storm overflow (OV) into the S1 channel, which transports the stormwater directly to the Silnica river (without treatment).

At a 30 m distance from the inlet of the main interceptor to the DC, the flow meter MES1 is installed, which measures the flow

rates during heavy rainfall with resolution of 1 minute. The probe of the flow meter measures the level (by measuring the water

level pressure) and average flow rate of the stormwater (via the Doppler effect), which, with the specific shape and dimensions

of the canal, calculates the volumetric flow rate of the stormwater (by means of a built-in microprocessor). Analysis of data

from 2010–2020 showed that during dry periods the measured flow rates varied between 1–9 dm³·s⁻¹, which indicates that





infiltration occurs in the stormwater network. Inspections of stormwater network operating carried out in the years 2008–2019

indicated that stormwater flooding occurs in the analysed catchment.

At a distance of 2.5 km from the catchment boundary, a rainfall measurement station is located, which provides constant measurement of rainfall with a 1-minute temporal resolution. Rainfall has been measured since 2008. Within the scope of the current study, the analysed catchment was divided into 8 sub-catchments (Fig. S1). In Tab. 1 the characteristics of sub-catchments are presented.


**Table. 1. Characteristics of sub-catchments**

| No. | F | Imp | Vk | Gk | R.t. | Vkp | dH1 | dHp | Lk | Jkp | Hst | Impd | Gkd | Vrd | Vkd |
|-----|-----|------|--------|-------------------|------|------|------|------|------|--------|------|------|-------------------|--------|--------|
| | ha | - | m$^3$ | m·ha$^{-1}$ | m | m$^3$ | m | m | m | - | m | - | m·ha$^{-1}$ | m$^3$ | m$^3$ |
| J | 12.66 | 0.37 | 157.0 | 0.0079 | 1.74 | 33.2 | 0.24 | 0.25 | 96.5 | 0.0036 | 1.42 | 0.40 | 0.0072 | 2159.4 | 2577.2 |
| K | 18.92 | 0.38 | 360.4 | 0.0084 | 1.69 | 28.4 | 0.31 | 1.05 | 56.5 | 0.0055 | 2.36 | 0.40 | 0.0063 | 1886.8 | 2373.7 |
| L | 27.15 | 0.36 | 557.4 | 0.0074 | 2.74 | 29.6 | 0.34 | 1.75 | 59.0 | 0.0058 | 2.36 | 0.42 | 0.0053 | 1496.0 | 2176.7 |
| M | 29.78 | 0.36 | 678.8 | 0.0071 | 4.49 | 48.7 | 0.38 | 1.15 | 62.0 | 0.0061 | 2.32 | 0.43 | 0.0050 | 1373.3 | 2055.3 |
| N | 36.78 | 0.37 | 712.2 | 0.0081 | 4.49 | 48.7 | 0.38 | 1.15 | 62.0 | 0.0061 | 2.32 | 0.44 | 0.0040 | 1061.4 | 2022.0 |
| O | 41.31 | 0.34 | 858.2 | 0.0079 | 5.32 | 16.1 | 0.21 | 1.28 | 20.5 | 0.0102 | 2.31 | 0.49 | 0.0037 | 825.9 | 1876.0 |
| P | 45.42 | 0.37 | 981.9 | 0.0082 | 5.64 | 16.1 | 0.21 | 1.28 | 20.5 | 0.0102 | 2.31 | 0.46 | 0.0027 | 682.2 | 1752.3 |
| R | 48.31 | 0.37 | 981.9 | 0.0088 | 5.64 | 16.1 | 0.21 | 1.28 | 20.5 | 0.0102 | 2.31 | 0.47 | 0.0023 | 553.1 | 1752.3 |
| S | 55.41 | 0.39 | 1240.2 | 0.0092 | 8.47 | 67.5 | 0.67 | 1.8 | 86.0 | 0.0078 | 2.31 | 0.55 | 0.0011 | 258.4 | 1493.9 |

where: F – catchment surface area; Imp – impervious area; Vk – volume of stormwater channel; Gk – length of stormwater channel per impervious area of the catchment; R.t. – height difference of the channel, Vkp – volume of the channel above the

closing cross-section of a catchment; dH1 – height difference of the terrain at section above closing cross-section r; dHp – height difference at section above closing cross-section; Lk – length of channel above closing cross-section of a catchment; Jkp – channel slope above closing cross-section of a catchment; Hst – the height of a manhole at closing cross-section; Imp – impervious area of downstream area; Gkd – length of a channel per impervious area below closing cross-section; Vrd – catchment retention above the closing cross-section calculated as $Vrd = F \cdot (Imp \cdot d_{imp} + (1-Imp) \cdot d_{per})$, Vkd – total retention of a

catchment.

**3 Methodology**

An innovative algorithm for creating a simulator to identify a specific flood volume was proposed (Fig. 2). In contrast to existing algorithms in use, in the adopted approach, a wider scope of computation was applied, linked with analysis of model sensitivity and the impact of uncertainty of calibrated SWMM model parameters on the probability of a stormwater network

failure. A failure was defined as exceedance of certain specific flood volume which points out that modernisation of the





stormwater network is necessary. In the proposed model, the value of a unit hydraulic overflow was defined as stormwater overflow per unit impervious area, which can be expressed by the following formula (Sinekamp and Pinekamp, 2011):

$$\kappa = \frac{\sum_{i=1}^{K} V_{t(i)}}{A_{imp}} \tag{1}$$

where: $V_t$ – volume of stormwater flooding from i-th manhole of the stormwater network, $K$ – number of manholes, $A_{imp}$ – impervious area.

The proposed computation algorithm consists of 11 modules. Module (1) provides data for developing a hydrodynamic model of a catchment, such as catchment characteristics (1a) – spatial development, slope of the terrain etc., stormwater network characteristics (1b) – diameters, lengths, channel slopes, manhole ordinates, etc., and measurement data regarding rainfalls and flow rates for calibration (1c).

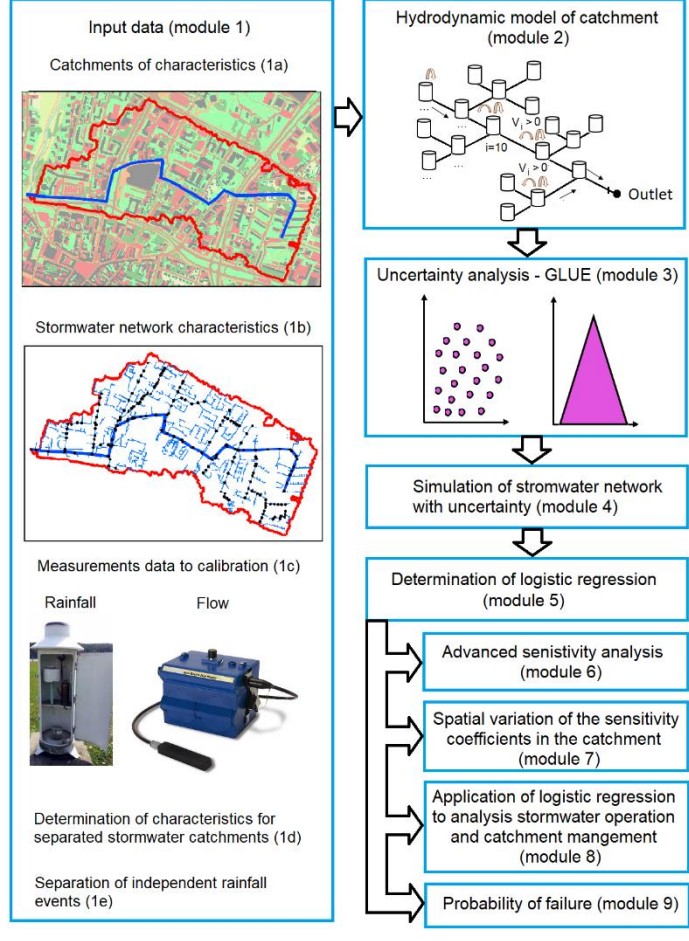

**Figure. 2. Algorithm for developing an advanced model to simulate a specific flood volume (situation maps in module (1a), (1b) by Walek (2019).**





In this module, the catchment is divided into sub-catchments along the main intercepting stormwater channel and the
characteristics of sub-catchments are defined (1d). Inside module (1) the long-term rainfall series are also included and
subsequently separated into independent rainfall events (1e). The data from module (1e) are used to develop a mathematical
SWMM catchment model (module 2). In module (3) an uncertainty analysis is performed using the GLUE method, basing on
hydrodynamic model of a catchment and rainfall – runoff data. The distribution of calibrated SWMM model parameters is
defined *a priori* and their values are calculated according to the Monte Carlo (MC) method. Simulations of a catchment with
mathematical model are performed with regard to uncertainty, while *a posteriori* distribution of parameters are defined
depending on measurement data from rainfall-runoff events. In the next stage (module 4) simulations of stormwater network
performance are computed for independent rainfall events determined in module (1e) and the values of unit hydraulic overflows
for sub-catchments determined in module (1d). Based on the simulation results and the assumption that when unit hydraulic
overflow exceeds 13 $m^3 \cdot ha^{-1}$ the continuous values are transformed to zero-one, a logistic regression model is developed
(module 5). This model is subsequently used to determine the sensitivity coefficients for calibrated SWMM parameters with
regard to rainfall intensity and catchment characteristics (module 6). Using adopted rainfall data, the sensitivity coefficients of
SWMM model parameters for sub-catchments are computed and maps showing sensitivity changes in catchment scale are
drawn (module 7). While the model is applied to identify stormwater flooding, the possible methods for improving stormwater
network operating are analysed inside module 8.

175          Using a logit model (module 5) and the uncertainty results from calibrated SWMM model parameters, the probability
of a specific flood volume is computed. Moreover, this module also analyses the impact of modifications of calibrated SWMM
model parameters on operating of stormwater network.

### 3.1. Determination of independent rainfall events (module 1e)

180          Determination of independent rainfall events for the period 2010-2019 was based upon criteria defined in DWA A-
118 (2006) guidelines. The minimum time period between independent rainfall events was set as 4.0 hours. Computation of
stormwater flooding was performed for rainfall events with a minimum depth of $P_t = 5.0$ mm (Fu and Butler, 2014) and only
for those events that resulted from convection rainfalls (i.e. rainfall duration below 120 min). Selection of rainfall events for
modelling stormwater flooding is a difficult task, usually requiring individual analysis (Schmidt 1997; Cea and Fraga, 2018;
Rahimi et al., 2021). For the analysed catchment, it was indicated that stormwater flooding occurs for C = 2, 3, 5 and rainfall
duration $t_r = 120$ min (Szeląg et al., 2021). The computed values of specific flood volume (the upper limit of 95% confidence
interval) are $\lambda = 45$ $m^3 \cdot ha^{-1}$.

          Based on analyses of the rainfall data for the period 2010–2018, it was observed that the number of rainfall events
with depths of $P_t = 5.2$–42 mm ranged from 12 to 30 in each year (202 rainfall events altogether). The maximum 30-minute
rainfall depths observed in the event of rainfall in any year were equal to $P_{t=30} = 2.5 - 25.6$ mm, while the rainfall durations
were between $t_r = 15 - 120$ min.



### 3.2. Hydrodynamic catchment model (module 2)

A SWMM program was used to determine a hydrodynamic model for the analysed catchment. To calculate manhole
overflow, the "Flooding" function was applied. For each calculation node K the inflow in time Q(t) was determined and the
volume of flooding was computed as $V_K = \int Q(t)_{F,i} dt$. On this basis, for sub-catchments i = 1, 2, 3, …, N, the total volume of
stormwater flooding was calculated and for the settled number of manholes (j) the specific flood volume ($\kappa_i$) was determined.

The model of analysed catchment covers 62 ha and is divided into 92 sub-catchments with areas varying from 0.12
to 2.10 ha and impervious areas ranging 5 to 95%. The model comprises 82 nodes and 72 sections of channels. At the
calibration stage, the retention of impervious areas was established as $d_{imp}$=2.50 mm and retention of permeable areas as $d_{per}$ =
6.0mm. Furthermore, it was assumed that the Manning roughness coefficient for impervious areas would be $n_{imp}$=0.025 m$^{-1/3}$·s, and $n_{per}$=0.10 m$^{-1/3}$·s for permeable areas. The flow path width was determined using the formula W=$\alpha$·A$^{0.50}$, where: $\alpha$ =
1.35. Catchment model calibration performed by Szeląg et al. (2021) indicated that for 6 rainfall-runoff events, a very good fit
of modelling outflow hydrographs to measurement results was obtained (Nash - Sutcliff coefficient was 0.85 - 0.98, coefficient
of determination was equal to 0.85 - 0.99, hydrograph volumes and maximum flows did not exceed 5% compared to
measurement data).

### 3.3. Uncertainty analysis – GLUE (module 3)

In the GLUE (Generalized Likelihood Uncertainty Estimation) method, the identification of model parameters was
considered as a probabilistic task due to the large number of parameters characterizing processes occurring in urban catchments
(runoff, infiltration, flow in stormwater conduits, flooding). The problem of parameter identification in the GLUE method is
formulated in the form of the Bayesian estimation relation (Beven and Binley, 1992):

$$P(Q/\theta) = \frac{L(Q/\theta)P(\theta)}{\int L(Q/\theta)P(\theta)} \tag{2}$$

where $P(\theta)$ stands for *a priori* (Tab. S1) parameter distribution; the *a priori distribution* of SWMM parameters represents the
initial assumption of parameter variability. In the case of mathematical models used to describe surface runoff, usually there
is no knowledge of the structure of its distribution and the range of acceptable parameter values resulting from their physical
interpretation is known at most. In the analysed case it was assumed that the distribution has uniform character (Dotto et al.,
2014; Knighton et al., 2016). The identification of model parameters in the GLUE method depends on the transformation of
an *a priori distribution* to an *a posteriori distribution* by means of a likelihood function ($L(Q/\theta)$), which determines the
probability of a combination of parameters depending on the quality of fit of the calculation result to the measured values. In
the present discussion the following form of the likelihood function was used (Romanowicz and Beven, 2000):

$$L(Q/\theta) = exp\left(\frac{r_t}{\varepsilon \cdot V(r_t)}\right) \tag{3}$$

$V(\cdot)$ – variance, $r_t$ - mean of the sum of squares of deviations of simulated value from measured value calculated as
$r_t = \frac{1}{l} \cdot \sum_{z=1}^{l} (Q_o - \widehat{Q}_i)^2$ (where: $Q_o$ and $\widehat{Q}_i$ denote z-th value from the times series of observed and computed flows; $\varepsilon$ is a
scaling factor for the variance of model residua, used to adjust the width of the confidence intervals. In Kiczko et al. (2018)





study, the value of $\varepsilon$ was determined, ensuring that 95% of observed discharge points is enclosed by 95% confidence intervals of the model output. Equation (2) is solved using the Monte Carlo method. In the first step, a sample of parameters is developed from an assumed *a priori* distribution. The model (SWMM in this case) is run with each combination of SWMM model parameters (Tab. S1) and from the calculated and measured outflow hydrographs (30 May 2010 and 8 July 2011) the values of
the likelihood function and *a posteriori* distributions are determined. Verification was performed based on the 15 September 2010 event and 30 July 2010 events. Detailed information on the obtained fit can be found in Kiczko et al. (2018). The likelihood measures and *a posteriori* distributions were computed using simulations performed for observed hyetographs and catchment outflow hydrographs (Szeląg et al. 2021).

### 3.4. Simulation of stormwater network operating with regards to uncertainty (module 4)

Based on the results of Monte Carlo simulation of calibrated SWMM model parameters (Tab. 1), the computation of stormwater network (5000 samples) was performed for separate independent rainfall events from the time period 2010-2019. This gave a total of 200·5000·9 = 9.000.000 rainfall - runoff events to develop a logistic regression model. The values of specific flood volume for sub-catchments (J, K, L, M, N, O, P, R, S) were calculated and zero-one variables were established
to develop logistic regression model. For computed values of specific flood volume ($\kappa \geq 13$ $m^3 \cdot ha^{-1}$) the variable value was denoted as 1, while in the opposite case it was 0 (Siekmann and Pinekamp, 2011).

### 3.5. Developing a logistic regression model (module 5)

Logistic regression, also known as the logit model, is a tool used for classification. Its benefits in comparison to widely
used methods arise from the fact that computation results range from 0 to 1 and they represent probability values. This model has been already applied for modelling storm overflows (Szeląg et al., 2020), identifying stormwater flooding from manholes (Jato – Espino et al., 2018) and the technical condition of sewage systems (Salman and Salem, 2012). The logistic regression model is described by the following equation:

$$p_m = \frac{\exp(\alpha_0 + \alpha_1 \cdot x_1 + \alpha_2 \cdot x_2 + \alpha_3 \cdot x_3 + \cdots + \alpha_i \cdot x_i)}{1 + \exp(\alpha_0 + \alpha_1 \cdot x_1 + \alpha_2 \cdot x_2 + \alpha_3 \cdot x_3 + \cdots + \alpha_i \cdot x_i)} = \frac{\exp(X)}{1 + \exp(X)} \tag{4}$$

where $p_m$ – probability of a specific flood volume (understood as the need to modernise the stormwater network); $\alpha_0$ – absolute term; $\alpha_1$, $\alpha_2$, $\alpha_3$, $\alpha_i$ – values of coefficients estimated with the maximum likelihood method, X – vector describing the linear combination of the independent variables; $x_i$ – independent variables describing rainfall characteristics, e.g., rainfall depth, its duration, and the parameters calibrated in the SWMM, catchment characteristics (permeability, terrain retention, density of stormwater network, length, slope, retention in stormwater channels etc.). Independent variables in the logit model were
calculated using the forward stepwise algorithm, recommended for the creation of such models. At the same time, it also ensures the elimination of correlated independent variables (Harrell 2001). The estimation of the coefficients $\alpha_i$ in equation (4) and thus the determination of the logistic regression model involved two stages: learning (80% of the data i.e. 7.200.000) and testing (20% of the data i.e. 1.800.000).





At the computation stage, the goal was to find such a value of threshold cutoff which would provide maximum fit of simulation
to measurement data. Thus, the subsequent cutoff values $p_m$ were tested until the best fit of measurement data and computation
results was obtained (SENS, SPEC → max of value). The fit of the calculation results to measurements was evaluated with
the following measures: sensitivity (SENS – determines correctness of classification in a set when the threshold values are
exceeded), specificity (SPEC – determines correctness of classification in a set when the threshold values are not exceeded)
and accuracy (Acc), which were discussed in detail in Harrell (2001). Equations for determining measures of fit between
computational results and measurements are provided in Supplementary Materials (Section 1).

### 3.6. Sensitivity analysis (module 6)

According to literature data (Morio, 2011), despite simplifications, local sensitivity analysis is widely applied at the
calibration stage and while analysing the hydrodynamic catchment models. In our study, the sensitivity coefficient was
calculated from the equation:

$$S_{xi} = \frac{\partial p_m}{\partial x_i} \cdot \frac{x_i}{p_m} \tag{5}$$

Where, knowing that $\frac{\partial p_m}{\partial x_i} = \beta_i \cdot p_m \cdot (1 - p_m)$, after transformations, the following formula was obtained:

$$S_{xi} = \beta_i \cdot x_i \cdot (1 - p_m) \tag{6}$$

Based on equation (6) it was concluded that the obtained value of sensitivity coefficient depends on probability $p_m$ and the
calibrated parameter of the SWMM model ($x_i$). Therefore, the above formula calculates $S_{xi}$ values for adopted rainfall durations
as well as catchment and SWMM model parameters.

In these considerations, $S_{xi}$ was calculated for calibrated SWMM model parameters (Tab. 2), at the same time
analysing the impact of rainfall duration ($t_r = 30 – 90$ min) for rainfall depth $P_t = 10$ mm (representative value for analysing
stormwater network functioning according to DWA – A 118, corresponding to an heavy rainfall event). For the above
assumptions, $S_{xi}$ was determined for different catchment characteristics, which at the same time helped to evaluate the
interactions between rainfall data and the parameter SWMM.

### 3.6.1. Spatial variability of sensitivity coefficients (module 7)

The probability of the specific flood volume ($p_m$) was computed using the logistic regression model for the catchment
characteristics defined in Tab. 2 and SWMM parameters established during calibration (Szeląg et al., 2016): for maximum
convection rainfall intensity for $t_r = 30$ min and $P_t = 9.62$ mm for Kielce (Section 2 at Supplementary Materials). The calculations
of Szeląg et al. (2022) proved that in the urban catchment in question there is a hydraulic overload of the stormwater system
due to convective rainfall. At the same time, the sensitivity coefficients for calibrated SWMM model parameters were
calculated. On this basis the spatial variability of $S_{xi}$ for the sub-basins was determined.




### 3.7. Application of the logit model to analyse stormwater operating and catchment management (module 8)

If the stormwater network ceases to function properly and the threshold value of $p_m$ is exceeded (please refer to section

3.6.1), some possible improvements were suggested, including: (a) increasing the retention depth of impervious areas, i.e. an increase of $d_{imp}$ from 2.50 mm to 3.50 mm, and at the same time raising the Manning roughness coefficient from $n_{imp} = 0.025$ $m^{-1/3}\cdot s$ to $n_{imp} = 0.035$ $m^{-1/3}\cdot s$, (b) an increase of hydraulic conductivity by reducing the Manning roughness coefficient for stormwater channels from $n_{sew}=0.018$ $m^{-1/3}\cdot s$ to $n_{sew}=0.012$ $m^{-1/3}\cdot s$. In addition, the possible change of spatial development of urban catchment area was taken into consideration. Finally, combinations of the above-mentioned computation variants were

analysed. When the values of independent variables (catchment characteristics) adopted for the calculations exceeded the lower/upper (e.g. for Imp = 0.34 - 0.39) limit of applicability of the determined logit model, the simulation results were verified with the hydrodynamic model. The verification procedure consisted of three stages:

a) computation of the probability of specific flood volume for rainfall with durations in the range of $t_r = 30–90$ min to assess stormwater network operating,

b) simulation with a calibrated hydrodynamic model for rainfall data as in stage (a),

c) comparison of computation results obtained in stages (a), (b); in the event of a of good fit, i.e. proper identification of specific flood volume, the results obtained from the logit model can be accepted.

### 3.8. Probability of stormwater network failure (module 9)

The probability of failure occurrence was used to assess stormwater network operating and the impact of uncertainty in SWMM model parameters in the event of rainfall. The failure was defined as a situation when the threshold value of specific flood volume ($\kappa = 13$ $m^3\cdot ha^{-1}$) is exceeded, also meaning that the stormwater network needs to be modernised. The probability of breakdown was calculated from the equation:

$$p_F = \frac{\sum_{j=1}^{N} Z_j}{N}, \ where: Z_j = \begin{cases} 1; \ p_m \geq p_{m,cr} \\ 0; \ p_m < p_{m,cr} \end{cases} \tag{7}$$

where: $p_m$ – probability of specific flood volume (exceedance of this value indicates a failure), $p_F$ – probability of the stormwater network failure in the event of rainfall, $Z_j$ – function describing stormwater network operation, for $Z_j = 1$ – drainage system requires modernisation; otherwise, i.e. $Z_j = 0$ – modernisation is not necessary.

For assumed value of characteristics of catchment, stormwater system and rainfall data ($x_i$ in the logistic model), the probability of a failure was computed. The impact of adjusting the Manning roughness coefficient for channels ($n_{sew}$) on $p_F$ was

considered. It was performed by introducing a threshold of maximum permissible values for the roughness of sewer pipes. It was assumed, that exceeding this value triggers modernisation of the sewer pipes with a new roughness. The above calculations were reduced to the following steps:

a) MC simulation of calibrated SWMM model parameters (N = 5000 samples),

b) computation of probability of specific flood volume for N items and establishment of failure probability,

c) computation of the Manning roughness coefficient for channels when $p_m > p_{m,cr}$ from the following formula:





$$n_{sew} = \frac{1}{\alpha_{nsew}} \cdot \left[ ln\left(\frac{p_{m,cr}}{1-p_{m,cr}}\right) - \left(\sum_{k=1}^{m-1} \alpha_k \cdot x_k\right) - \left(\sum_{n=1}^{p} \alpha_n \cdot x_n\right) - \left(\sum_{s=1}^{t} \alpha_s \cdot x_s\right) \right] \qquad (8)$$

where: k = 1, 2, 3, …, m – calibrated SWMM model parameters; n = 1, 2, 3, …, p – catchment characteristics; s = 1, 2, 3, …, t – rainfall characteristics, $\alpha_{nsew}$ – estimated coefficient in logistic regression model for the Manning roughness coefficient for channels.

d) establishment of empirical distribution describing the $n_{sew}$ values calculated from Equation (8),

e) computation of $n_{sew}$ values from Equation (8) for $n_{sew(un)} \leq n_{sew(m)}$ (where: $n_{sew(un)}$ – Manning roughness coefficients of channels computed in step (a), $n_{sew(m)}$ – maximal boundary (threshold) value of Manning roughness coefficient for channels), when $n_{sew(un)} \geq$ $n_{sew(m)}$                                                                                                              to $n_{sew} = n_{sew(un)}$,

f) computation of probability of specific flood volume and probability of failure,

g) determination of empirical distribution (CDF) for $n_{sew}$,

h) steps e – g are repeated r = 1, 2, 3, .., z – for different values of $n_{sew,max}$ and median values of $n_{sew(0.5)} = f(n_{sew(m)}, r)$ are denoted based on empirical distributions,

i) steps a–h are conducted for different catchment characteristics,

j) graph $p_F = f(n_{sew(0.5)})$ is drawn.


## 4. Results

### 4.1. Uncertainty analysis – GLUE (module 3)

Based on SWMM simulation results including uncertainty of calibrated parameters (Tab. S1), the likelihood functions were determined (Kiczko et al., 2018). The form factor was ε= 65, ensuring that the 95% confidence intervals contain 95% of the
observational data used for calibration. The approach adopted enabled the estimation of the predictive uncertainty and not just the parametric uncertainty of the model. For the observational events (30 May 2010 and 8 July 2011) used to identify the SWMM parameters, it was found that 96% of the measurement points included the calculated confidence interval. For the validation sets, 90% of the observation points fall within the bands for the 15 September 2010 event and 70% for 30 July 2010. The results of the likelihood function calculations for the calibrated SWMM model parameters are given in Figs. S3 - S10 in Supplementary
Information.

### 4.2. Simulations of stormwater network operation with regard to uncertainty (module 4)

Based on the results of calculations of stormwater system performance (for separated rainfall events) and taking into account uncertainties, the variation of specific flood volume for the separated sub-catchments was determined; the results are
presented in Fig. 3. Based on the obtained curves it can be stated that the uncertainty of SWMM parameters influences the simulation results, which is confirmed by the great variability of the 1% and 99% percentile values for each sub-catchment. By analysing the median values, it was discovered that the largest specific flood volume was obtained for sub-catchment J (14.90 m³·ha⁻¹), and 8.29 m³·ha⁻¹ for the smallest sub-catchment. The simulation results for the 1% percentiles show that for adopted rainfall events ($P_t > 5.0$mm and $t_r < 150$ min) stormwater flooding occurred in sub-catchments.





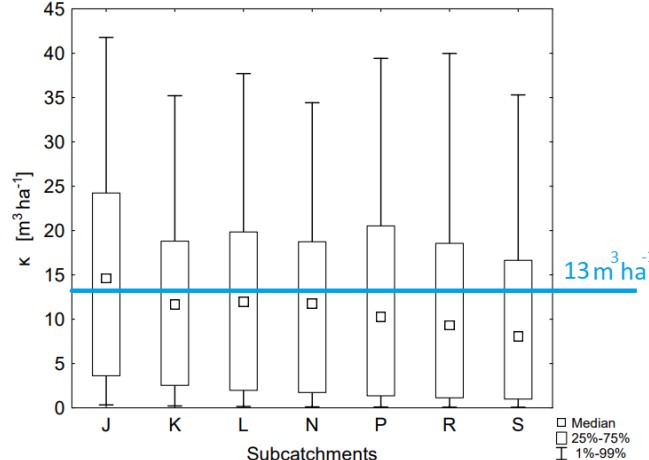

**Figure. 3. Variability of specific flood volume for sub-catchments.**

By analysing the range of variability for determined percentiles, it was demonstrated that problems with operating of the stormwater network are present in each sub-catchment, since the calculated values of percentiles (75%, 99%) are higher than 13 m$^3$·ha$^{-1}$. This indicates that the stormwater network requires modernisation.

### 4.3. Determination of the logistic regression model (module 5)

A logistic regression model was built based on the operational simulation of the stormwater network. The model can be used to identify specific flood volume and for decision-making regarding modernisation of the stormwater system. The relationship from Equation (3) can be described by the following linear combination:

$$X = X_{rain} + X_{SWMM} + X_{Catchm} - 54.15 \tag{9}$$

$$X_{rain} = 4.05 \cdot P_{tot} - 0.18 \cdot t_r \tag{10}$$

$$X_{SWMM} = 0.23 \cdot \alpha - 79.40 \cdot n_{imp} + 6.23 \cdot \beta + 0.33 \cdot \gamma + 234.12 \cdot n_{sew} \tag{11}$$

$$X_{Catchm} = 76.72 \cdot Imp + 40.77 \cdot Impd - 0.01 \cdot Vk - 1967.04 \cdot Gk - 1169.00 \cdot Gkd - 20.33 \cdot Jkp \tag{12}$$

For other independent variables (Tab. S2) the determined coefficients appeared to be statistically insignificant in prediction confidence band 0.05. Standard deviations of the coefficients estimated from the logit model and the test probabilities are presented in Tab. S2. The performed computations indicated that the best fit of the computed results to the measurement data was obtained for $p_{m,cr} = 0.75$. For the test data set (20%) the following values were obtained: SPEC = 95.24%, SENS = 84.62% and Acc = 87.87%.

### 4.4. Sensitivity analyses (module 6)

For rainfall depth $P_{tot} = 10$ mm and duration $t_t = 30 - 90$ min, the sensitivity coefficients for the SWMM model were determined, based on Equation (6). For calculation of $S_{xi}$ the values established during calibration were adopted (Kiczko et al., 2018).





The computation results for two parameters of the SWMM model ($\beta$ and $n_{sew}$,) are presented in Fig. 3. These two parameters appeared

to have the most significant impact on specific flood volume and, at the same time, they present a vastly different impact on the

dynamics of changes regarding $S_{xi} = f(t_r, Imp, Impd, Vk, Jkp)$; the calculation results for the other SWMM model parameters are

given in Figs. S11–S15 (Supplementary Information).

Analysis of the curves presented in Fig. 4 and Figs. S11 – S15 indicates that for the adopted values of $t_r$ and Imp, Impd, Vk,

Jkp, the highest values of $S_{xi}$ was obtained for correction coefficient percentage of impervious areas ($\beta$), Manning roughness

coefficient for sewer channels ($n_{sew}$) and Manning roughness coefficient for impervious areas ($n_{imp}$). Among the analysed

SWMM model parameters, the retention depth of impervious areas ($d_{imp}$) had the lowest impact on the results of hydraulic

overflow computation. Based on the determined curves (Fig. 4), it was concluded that rainfall duration had a significant impact

on the computation results for sensitivity coefficients. It was found that elongation of rainfall duration time ($t_r$) results in an

increase of absolute $S_{xi}$ values.

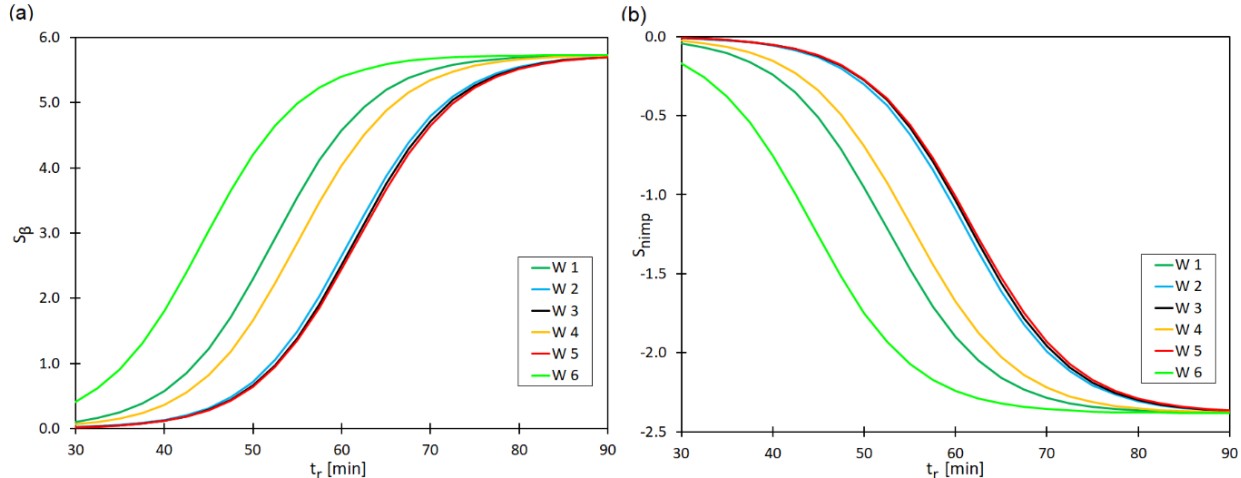


**Figure 4. The impact of rainfall duration ($t_r$) and catchment characteristics (Imp, Impd, Vk, Jkp) on sensitivity coefficients: (a) $S_\beta$, (b) $S_{nimp}$ (where: variants W1, W2, W3, W4, W5, W6 according to Tab. 2).**

**Table 2. Combinations of catchment characteristics adopted for analysis**


| Wariant | Imp | Impd | Gk | Jkp | Vk |
|---------|------|------|--------|-------|-----|
| W1 | 0.34 | 0.40 | 0.0075 | 0.010 | 400 |
| W2 | 0.36 | 0.40 | 0.0075 | 0.010 | 400 |
| W3 | 0.34 | 0.44 | 0.0075 | 0.010 | 400 |
| W4 | 0.36 | 0.40 | 0.0075 | 0.010 | 500 |
| W5 | 0.36 | 0.40 | 0.0075 | 0.004 | 400 |
| W6 | 0.36 | 0.40 | 0.0090 | 0.010 | 400 |

For instance, an increase of rainfall duration from $t_r = 30$min to $t_r = 90$min (W2, W3, W5) results in an increase of $S_\beta = 0.01$ to

$S_\beta = 5.70$ and a change of $S_{nimp} = -0.01$ to $S_{nimp} = 2.40$. It was also determined that an increase of Imp, Impd results in a decrease

of $S_\beta$ and $S_{nimp}$ sensitivity coefficients. For instance, an increase of Imp from 0.34 to 0.36 results in a decrease of $S_\beta$ from 1.23





to 0.28; identical values were obtained for Impd. Moreover, an increase of Vk, Jkp, Gk leads to an increase of $S_\beta$ and $S_{nimp}$ sensitivity coefficients. For example, when Vk increased from $400\text{m}^3$ to $500\ \text{m}^3$, $S_\beta$ increased from 0.29 to 0.82. Additionally, a 10% growth of $S_\beta$ was observed due to a change of Jkp = 0.004 to Jkp = 0.010. Finally, when Gk increased from 0.0075 to 0.009 $S_\beta$ also increased from 0.29 to 3.03.

**4.5. Spatial distribution of sensitivity coefficients (module 7)**

Based on the determined logit model described by Equation (9) and proposed relationship (6), the probabilities of specific flood volume for sub-catchments ($p_m$) were computed and sensitivity coefficients for calibrated SWMM parameters were determined (Tab. S1). A rainfall duration time of $t_r$ = 30 min was adopted. According to Szeląg et al. (2022) this rainfall duration results in a specific flood volume in the analysed stormwater network. Computation results regarding $p_m$, $S_\beta$, $S_{nsew}$,

$S_{nimp}$ values (which had the highest impact on specific flood volume among SWMM parameters) are presented in Fig. 5.

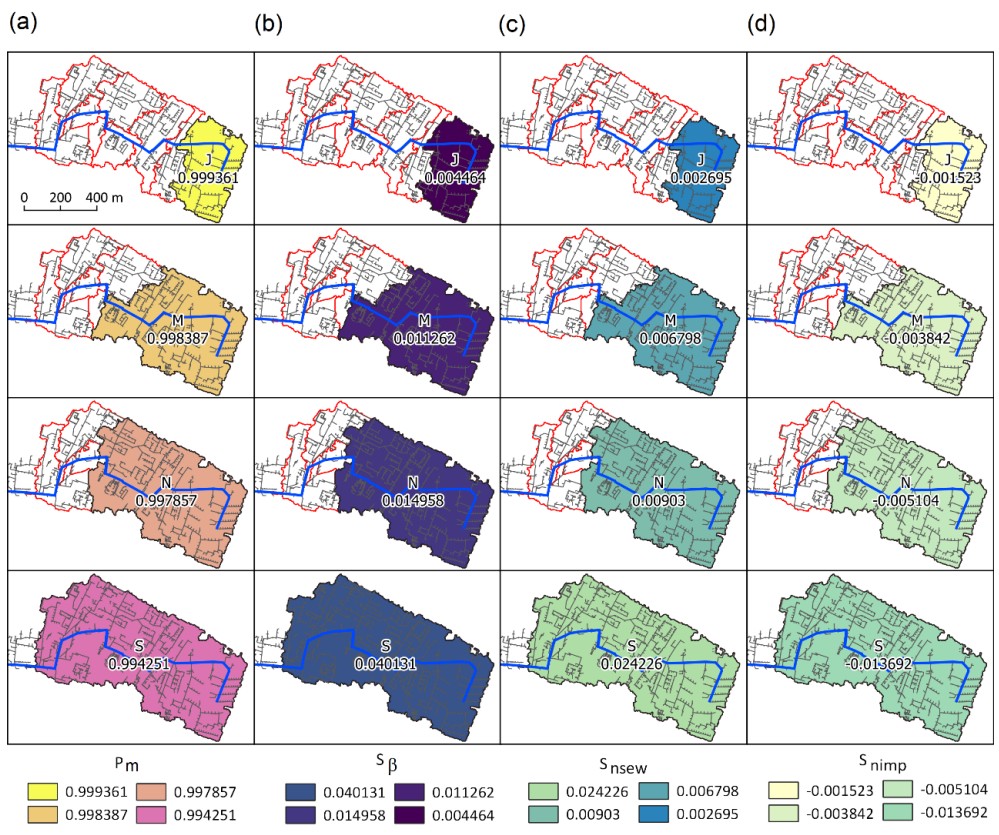

**Figure 5. (a) Probability of specific flood volume ($p_m$); (b) sensitivity coefficient $S_\beta$; (c) sensitivity coefficient $S_{nsew}$; (d) sensitivity coefficient $S_{nimp}$ in sub-catchments J, M, N, S.**

The probability of specific flood volume and other SWMM model parameters is presented in Figs. S16–S24. Based

on the conducted calculations it was found out that for adopted rainfall data (convection rainfall with a duration time of 30


minutes) the problems with proper operating of stormwater network appear, which is confirmed by the values of specific flood volume for subsequent sub-catchments (Fig. 5a). The obtained results for sensitivity coefficients $S_\beta$, $S_{nsew}$, $S_{nimp}$ confirm the results presented in section 4. Among the analysed SWMM parameters, the highest values of $S_{xi}$ were determined for correction coefficient for percentage area ($\beta$). The calculations indicated that the lowest values of sensitivity coefficients ($S_\beta$, $S_{nsew}$, $S_{nimp}$)

were obtained for catchment and the highest for catchment S.

### 4.6. Implementation of logit model to analyse the operating of the stormwater network and catchment management (module 8)

Due to the fact that in the analysed stormwater network an exceedance of specific flood volume was observed,
possible improvements to the network were considered in terms of correcting catchment imperviousness (Imp) as well as enhanced terrain retention and channel conductivity. In variant I imperviousness Imp was reduced by 10%. In variant II $d_{imp}$=3.5 mm and $n_{imp}$ =0.035 m$^{-1/3}\cdot$s were corrected. Variant III was a combination of variants I and II, where channel conductivity was also increased ($n_{sew}$=0.012 m$^{-1/3}\cdot$s). The results of $p_m$ computations are presented in Fig. 6, while Fig. 7 shows $S_\beta$ for variants I, II and III for sub-catchments. Simulation results for the sensitivity coefficients of other SWMM model
parameters (Tab. S1) and the probability of specific flood volumes are presented in Figs. S16–S24

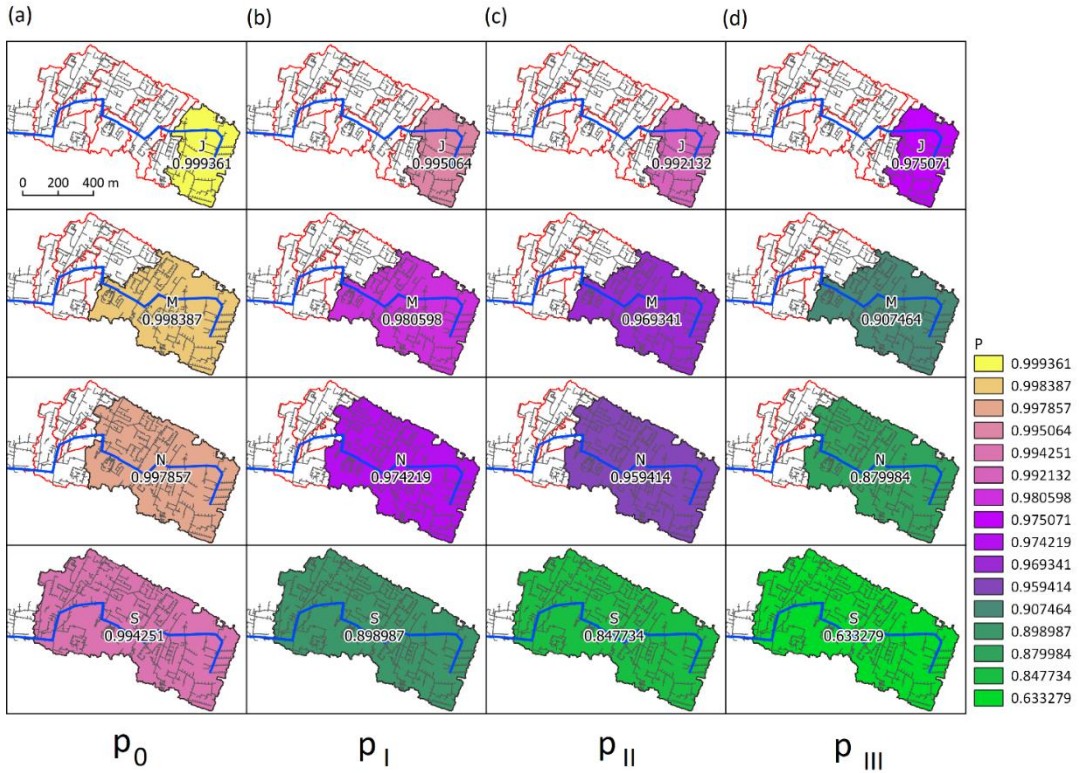

**Figure 6. Probability of specific flood volume in sub-catchments for: (a) present state ($p_0$) and for (b) I, (c) II, (d) III modernisation variants.**





The obtained results presented in Fig. 6 indicate that the reduction of sub-catchment imperviousness (Imp) improves operating
of the stormwater network. Among the analysed sub-catchments, the worst conditions for stormwater system operation were
found in sub-catchment J. Despite reducing imperviousness while also enhancing retention depth and increase of channel
conductivity, $p_m$ only decreased from 0.994 to 0.975. Regarding the whole analysed catchment (sub-catchment S) the reduction
in imperviousness by 10% resulted in a 10% decrease of specific flood volume probability. When terrain retention and channel

conductivity are increased alongside a reduction in imperviousness, the probability of specific flood volume $p_m$ decreases by
approximately 36%. After analysis of the change of $p_m$ values in sub-catchments J, M, N, S for modernisation variant III, it
was found out that despite enhancing retention depth and channel conductivity while reducing catchment imperviousness,
hydraulic overloads ($\kappa = 13$ m$^3$·ha$^{-1}$) still occur in the analysed catchment. This indicates the need for further changes in both
the catchment area and stormwater network. For variants I, III the Imp values for the sub-basin are below the applicability

range of the logit model, therefore in order to verify the results obtained hydrodynamic model simulations were performed
(Tab. 3S). The results of the model calculations confirm their high agreement; out of 72 cases, identical results were obtained
in 68. In addition to these calculations, the variability of sensitivity coefficients ($S_\beta$) in sub-catchments for modernization
variants I, II and III was also analysed (Fig. 7).

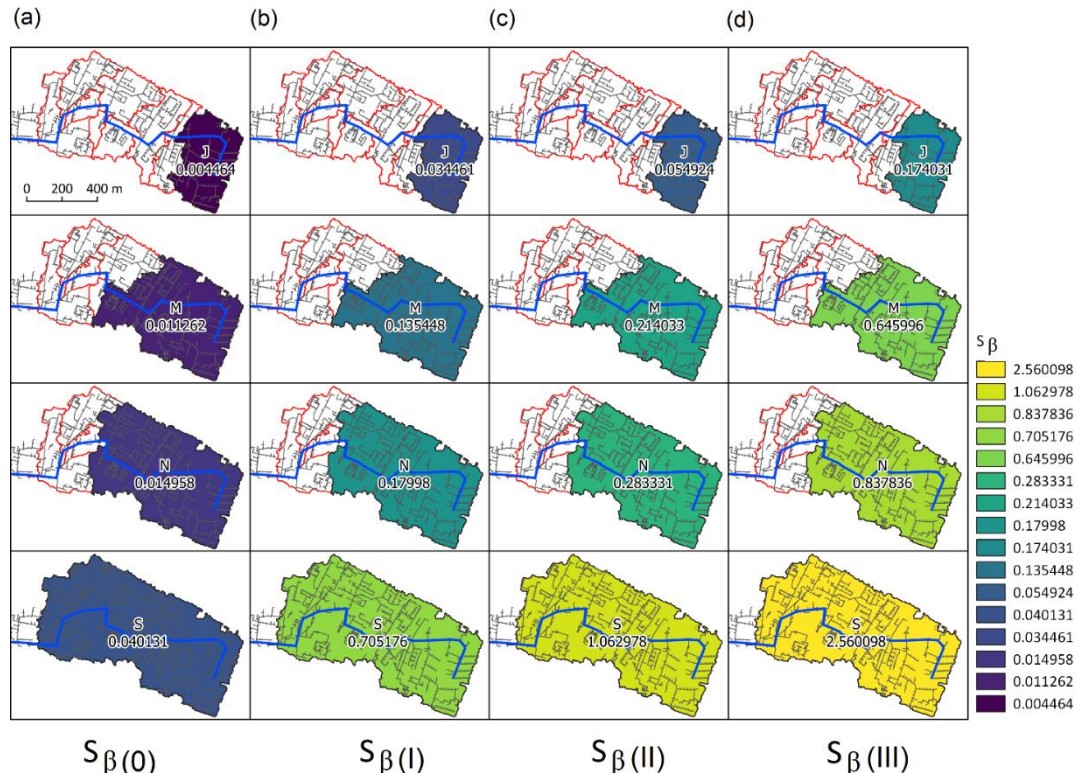

**Figure 7. Sensitivity coefficient ($S_\beta$) in sub-catchments for: (a) present state (0) and for (b) I, (c) II, (d) III modernisation variants.**





Based on the calculations performed, it was demonstrated that reducing catchment imperviousness (Imp) as well as increasing terrain retention and channel conductivity lead to higher sensitivity coefficient $S_\beta$ values in sub-catchments (Fig. 7). Moreover,

the relationships presented in Fig. 5 were confirmed, indicating that in subsequent closing cross-sections of sub-catchments J, M, N, S the sensitivity coefficient $S_\beta$ increases. For variant III it was indicated that $S_\beta$ for sub-catchment J was 0.174, and 2.561 for sub-catchment S. These results provide relevant information for planning retention infrastructure that reduces outflow – for instance, Green Infrastructure (GI) facilities. They also point out the need to widen the range of applicability of proposed simulator as well as for including parameters referring to GI, which would allow for planning of their location inside analysed

catchments.

## 4.7. Probability of failure (module 9)

Based on SWMM model parameters determined via the MC method (Tab. S1), probability of failure ($p_F$) was computed for convection rainfall in Kielce with a duration time of $t_r$=30 min and $P_{tot}$= 9.61 mm. The following threshold values

of $n_{sew(m)}$ were adopted for calculations: $n_{sew(m)}$ = 0.015; 0.02, 0.025; 0.030, 0.035, 0.040, 0.045 $m^{-1/3}·s$, coupled with three variants of catchment characteristics: Imp = 0.36 and Impd =0.40; Imp = 0.35 and Impd = 0.40; Imp = 0.35 and Impd = 0.42. The Manning roughness coefficients of the channels ($n_{sew}$) for the analysed variants were presented as empirical distribution (CDF). In Figure 8a the results for Imp = 0.36 and Impd = 0.40 are presented, while other variants are shown in Figs. S25, S26. Figure 8b presents the impact of $n_{sew}$=f($n_{sew(m)}$) for percentiles 0.25 and 0.50 (based on the curves in Fig. 8a, S25, S26 the

values of the respective percentiles for the analysed $n_{sew(m)}$) on the probability of failure ($p_F$).

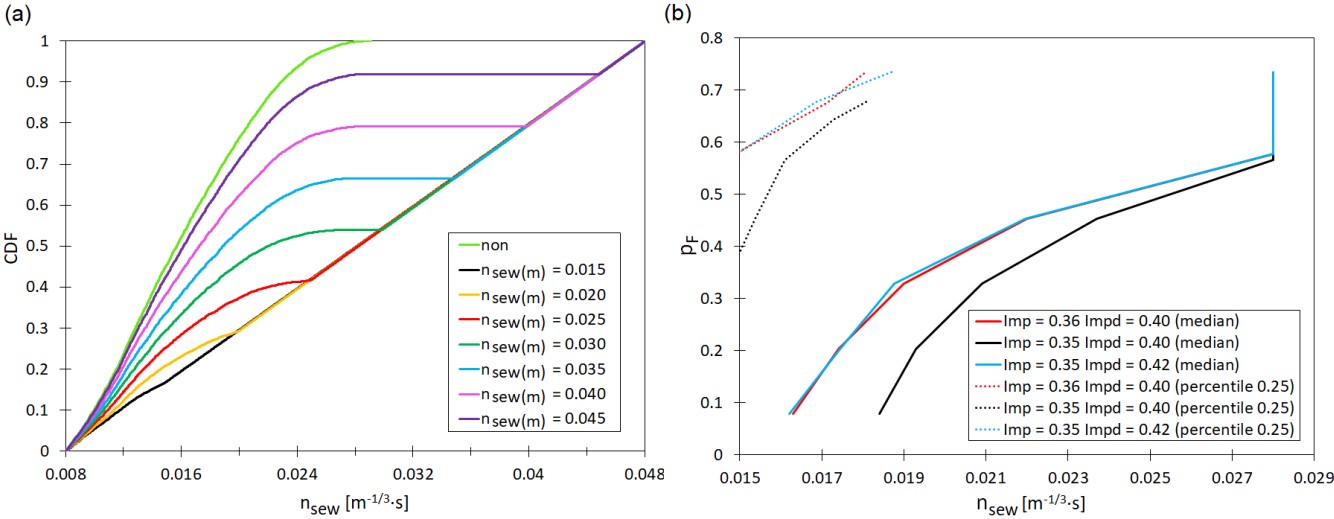

**Figure 8. (a) Empirical distributions of threshold values of Manning roughness coefficients of channels ($n_{sew}$). (b) Impact of Manning roughness coefficient for channel on failure probability ($p_F$) in relation to Imp, Impd.**






Assuming that Manning roughness coefficients – $n_{sew(un)}$ determined by MC simulation which exceeds the threshold triggers the modernisation of sewer pipes resulting in reduction of roughness below $n_{sew(m)}$ following the condition in which the stormwater network functions ($p_m = f(X_{rain}, X_{SWMM}, X_{Catchm}) > 0.75$ for an independent rainfall event), it was found out, that an appropriate decrease of percentiles (0.25 and 0.50 - median) leads to improved network operation and to a lower failure

probability (Figs. 8a, 8b). Based on computation results, it was observed that the change of percentile 0.50 for $n_{sew}$ for a sample from MC simulation leads to a decrease from 0.028 $m^{-1/3}\cdot s$ to 0.021 $m^{-1/3}\cdot s$ (as a result of correction $n_{sew(un)} < n_{sew(m)}$) and to improved stormwater network operation understood as a lower probability of failure (decrease of $p_F$ from 0.68 to 0.42 for Imp $= 0.36$ and Impd $= 0.40$). These results confirm the significance of catchment characteristics (Imp, Impd) for the operability of a stormwater network. For Impd $= 0.40$, the reduction in catchment impervious area (Imp) from 0.36 to 0.35, at percentile

$n_{sew} = 0.019$ $m^{-1/3}\cdot s$ results in a decrease in failure probability from $p_F = 0.42$ to $p_F = 0.33$.

## 5. Discussion

Developing and calibrating mathematical models to simulate stormwater network operation under hydraulic overloads is one of the latest areas of research. The analyses performed within our study identified two SWMM model parameters, which

had the highest impact on the unit hydraulic overload. These parameters were the correction coefficient of impervious area ($\beta$) and the Manning roughness coefficient for channels ($n_{sew}$). The parameters referring to retention in impervious areas had less impact. The results of our computations are consistent with the findings of Thorndahl et al. (2009), who applied the FORM (first order reliability model) method to simulate flooding from a single manhole in the Frejlev catchment (Belgium), based on rainfall data and calibrated parameters of a hydrodynamic model. In addition, Fraga et al. (2016) also indicated the relevance

of the Manning roughness coefficient for channels to calculate flooding in a small road catchment via the GLUE-GSA method. Similar results were obtained in simulation of hydraulic overload performed by Fu et al. (2012), who applied the calibrated hydrodynamic model of a catchment, where the uncertainty of calibrated parameters was evaluated by means of fuzzy logic.

The results of our study confirmed the major significance and huge interaction between catchment characteristics and SWMM model parameters. This fact can be further compared by several references (Li and Willems, 2020; Jato – Espino et

al., 2019; Zhuo et al., 2019) presenting comparisons of flooding simulations in urban catchments. However, to date, there has been no statistical model that would take into account both hydrodynamic model parameters as well as catchment and stormwater network characteristics. Our analysis indicated that an impervious area in a catchment (Imp, Impd) leads to the increase of flooding, while an increase in channel volume above the closing cross-section of a catchment (Vk) and its longitudinal slope (Jkp) results in the decrease of flooding. Interestingly, the increase of unit impervious area per the length of

main stormwater interceptor (Gk, Gkd) results in smaller volume of stormwater flooding. Jato – Espino et al. (2018), who developed a logistic regression model based on simulation with a hydrodynamic model, confirmed the significant impact of catchment imperviousness and longitudinal channel slope on specific flood volume. The impact of Gk, Gkd indicates that an increase in interceptor length leads to the increased probability of specific flood volume. This result is absolutely right due to the fact that the longer the channel, the greater the number of manholes. Therefore, the number of potential points where





stormwater flooding can occur also increases. This is dependent on conditions of stormwater network performance and local circumstances (Mignot et al., 2019). Due to the local character of flooding phenomena, further analyses should concentrate on other stormwater catchments.

In comparison to the statistical models used so far (Li and Willems, 2019; Thorndahl 2009, Ke et al., 2020), the approach proposed in our study includes SWMM model parameters describing catchment retention and, at the same time, the
characteristics of the catchment and stormwater network. For instance, Jato – Espino et al. (2018, 2019) and Li and Willems (2020) only included catchment characteristics for modelling hydraulic overflow. In the wider context of catchment management, their approach does not apply for the characteristic of the materials used for road, roofs or parking places, etc. On the other hand, Fu et al. (2012) and Thorndahl et al. (2009) focused on the uncertainty of identified parameters, which only helped to correct the retention of impervious areas, Manning roughness coefficient etc., without the possibility of changing
their surface area inside the catchment, which limits possible applications of these models for catchment management. The approach proposed in our study is a combination of these two solutions, which provides a tool which can be successfully implemented to manage other catchments.

Authors tackling topics related to stormwater flooding in urban catchments (Fraga et al., 2016; Razhavi and Gupta, 2015; Cristiano et al., 2017; Gao et al., 2020) stress the significance of proper identification of calibrated parameters of
hydrodynamic models and the growing role of sensitivity analysis. Such analyses are usually performed for single catchments with different characteristics and for rainfall data with different temporal pattern, which leads to the neglection of many important factors. The approach adopted in our study demonstrated that the values of sensitivity coefficients in SWMM model are influenced both by catchment characteristics and by rainfall intensity. In comparison to the methods currently used, this shows the importance of carefully selecting rainfall-runoff events at the calibration and validation stage. Due to the fact, that
the adopted solution includes catchment and stormwater network characteristics, it enables an evaluation of the spatial variability in SWMM model parameter sensitivity to computation results of stormwater flooding. Thus, it can be used as an assessment tool indicating a requirement for stormwater network modernisation. Furthermore, in contrast to other methods, the adopted approach enables a preliminary evaluation of the impact of calibrated parameters on simulation results, which can be of significance when locating measurement devices and for using adequate techniques to measure the parameters required
for model development (Cheng et al., 2021; Meng et al., 2019, Jato – Espino et al., 2016).

## 6. Conclusions

Modelling stormwater network operation under hydraulic overloads and model calibration is one of the latest research areas, as confirmed by a number of scientific studies. However, due to the complex dynamics of hydraulic processes in
stormwater networks, this issue requires further field studies and simulation experiments.

In this study, a computation method for creating an innovative simulator to analyse how stormwater networks operate and evaluate modernisation requirements is proposed, based on the specific flood volume. In the adopted solution, the impact of rainfall data, catchment characteristics (impervious areas in the lower and upper part) and stormwater network characteristics



(the length of channel per unit impervious area, channel slope and volume) as well as SWMM parameters were included

simultaneously. To our knowledge, no other previous study has included such a broad scope of analysis. Performed computations indicated that a logistic regression model can be successfully applied to simulate such a complex phenomenon as hydraulic overload in stormwater network, with regard to a broad set of variables. Due to the fact that catchment and stormwater network characteristics were included in the model, this calculation tool can be applied to other stormwater systems as well as to catchment management. In this context, the developed tool can be viewed as an alternative to hydrodynamic

models, which can be used at the preliminary stage of analyses related to spatial planning, urban development and expansion etc. This is of major significance since at the preliminary stage, the data set for building catchment models is limited, and, at the same time, there is an urgent demand for simulation tools to assist decision making. Such tools are needed for instance for assessment of the location of retention tanks within a stormwater network or implementation of Green Infrastructure solutions, which assure the assumed hydraulic effect at low economic cost, and, at the same time, meet the urban landscape planning

aspects, increase citizens' well-being and have a small environmental footprint.

The results of this study confirm the impact of catchment and stormwater network characteristics as well as SWMM parameters on the probability of specific flood volume. Furthermore, the obtained simulation results show the strong interaction between the above-listed parameters, previously not included in sensitivity analysis. Based on the obtained results it is concluded that rainfall intensity as well as catchment and stormwater network characteristics have a significant impact on the

computed values of sensitivity coefficients. This is extremely relevant in the context of calibration of models that can be applied to analyse stormwater network operation and to support the decision-making process (management of stormwater in an urban catchment). The obtained results indicate the need to carefully select rainfall-runoff events to calibrate and validate a model, with respect to rainfall intensity. Since the proposed solution analyses the spatial distribution of sensitivity coefficients, it is possible to identify the most vulnerable areas inside a catchment that require specific attention while

identifying SWMM model parameters, which could also be taken into account when locating measuring instruments.

Due to the wide applicability of the proposed simulators to analyse how urban stormwater networks operate, further verification in other urban catchments is required. Specific interest should be focused on spatial distribution of urban catchments which would enable stormwater networks to be designed alongside retention and Green Infrastructure utilities in order to minimize negative environmental impact caused by urbanisation and improve standards of living.


## 7 Appendices

**Appendix A: List of Symbols**

Symbols:

$A_{imp}$ – area of impervious surface (ha),

$dH1$ – height difference of the terrain at section above closing cross-section (m),

$dHp$ – height difference at section above closing cross-section (m),



$CDF$ – Cumulative Distribution Function (–),

$d_{imp}$ – retention depth of impervious areas (mm),

$d_{perv}$ – retention depth of pervious areas (mm),

$F$ – catchment surface area (ha),

$Gk$ – length of stormwater channel per impervious area in a catchment (m·ha$^{-1}$),

$Gkd$ – length of a channel per impervious area below closing cross-section (m·ha$^{-1}$),

$GLUE$ - Generalized Likelihood Uncertainty Estimation,

$Hst$ – the height of a manhole at closing cross-section (m),

$Imp$ – impervious area,

$Impd$ – impervious area of a catchment of downstream area,

$J$ – average rainfall intensity (l·(s·ha)$^{-1}$),

$Jkp$ – channel slope above closing cross-section of a catchment

$K$ – total number of sewer manholes (–),

$Lk$ – length of channel above closing cross-section of a catchment (m),

$L(Q/\theta)$ – likelihood function,

$n_{imp}$ – Manning roughness coefficient for impervious areas (m$^{-1/3}$·s),

$n_{perv}$ – Manning roughness coefficient for pervious areas (m$^{-1/3}$·s),

$n_{sew}$ – Manning roughness coefficients of sewer channels (m$^{-1/3}$·s),

$Q_z$ – denote $z$-th value from the times series of observed and computed discharges (m$^3$·s$^{-1}$),

$P_t$ – maximum depth of rainfall (mm),

$p$ – cumulative distribution function (CDF),

$p_m$ – probability of specific flood volume,

$P(\theta)$ – stands for *a priori* parameter distribution,

$R.t.$ – height difference of the channel (m),

$S_{xi}$ – sensitivity coefficient,

$x_i$ – independent variables,

$SWMM$ – Storm Water Management Model,

$t_r$ – duration of rainfall (min),

$V\,()$ – variance,

$Vk$ – volume of stormwater channel (m$^3$),

$Vkd$ – total retention of a catchment.

$Vkp$ – volume of the channel above the closing cross-section of a catchment (m$^3$),

$Vrd$ – catchment retention above the closing cross-section (m$^3$),

$V_{t(i)}$ – floodings volume from $i$ - th sewer manhole (where: $i$ = 1, 2, 3, …, k) (m$^3$),



$W$ – width of the runoff path in a subcatchment (m),

$\alpha$ – Coefficient for flow path width (–),

$\beta$ – Correction coefficient for percentage of impervious areas (–),

$\gamma$ – Correction coefficient for subcatchment slope (–),

$\varepsilon$- a scaling factor for the variance of model residua, used to adjust the width of the confidence intervals,

$\kappa$ – specific flood volume ($m^3 \cdot ha^{-1}$),

**Code availability:** The authors announce that there is no problem sharing the used model and codes upon request to the
corresponding author.

**Data avaiibility:** The authors confirm that data supporting the findings of this study are available from the corresponding
author upon request.

**Author contribution:** Conceptualization: Szeląg, Methodology: Fatone, Szeląg, Kiczko; Formal analysis and investigation:
Szeląg, Kiczko, Stachura, Wałek; Writing - original draft preparation: Szeląg, Wojciechowska, Wałek, Fatone; Writing -
review and editing: Wojciechowska, Fatone; Supervision: Szeląg, Wojciechowska

**Competing interests:** The authors declare that they have no conflicts of interest.

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
