# Peer review of "The role of catchment characteristics, sewer network, SWMM model parameters in urban catchment management based on stormwater flooding: modelling, sensitivity analysis, risk assessment"

_Hydrology and Earth System Sciences, 2022_

## Referee Comment (RC1)

**Ref**: HESS_2022_109_review (submitted on 25 April 2022)
**Title**: The role of catchment characteristics, sewer network, SWMM model parameters in urban catchment management based on stormwater flooding: modelling, sensitivity analysis, risk assessment

**Authors**: Bartosz Szeląg*, Adam Kiczko, Grzegorz Wałek, Ewa Wojciechowska, Michał Stachura5, Francesco Fatone

**Comments**

This manuscript presents a model to simulate specific flood volume considering both catchment and stormwater network characteristics, and including a module to calculate and indicate possible failure of the stormwater network, which is an interesting topic. The presented model algorithms consist of nine modules (Section 3 methodology): eight modules are a replication of what were developed and published by the same lead author; i.e., Szeląg et al. 2021 (e.g., hydrodynamic model- module 2 in the manuscript, sensitivity test considering uncertainty- modules 3, 4, 6, 7) and Szeląg et al. 2022 (Logistic regression and its application to stormwater network - modules 5, 8) over the same catchment. Module 1 in this manuscript addresses 9 sub-catchments (Table 1, though mentioned as 8 in line 128) to be simulated, which the items of characteristics are applied the same way as above-mentioned articles but the resulted values shown in Table 1 are slightly different depending on the differently selected sub-catchment.

> Szeląg, B., Kiczko, A., Łagód, G., De Paola, F.: Relationship between rainfall duration and sewer system performance measures within the context of uncertainty, Water Res Manage., 35, 5073 – 5087, https://doi.org/10.1007/s11269-021-02998- x, 2021
>
> Szeląg, B., Suligowski, R., De Paola, F., Siwicki, P., Majerek, D., Łagód, G.: Influence of urban catchment characteristics and 760 rainfall origins on the phenomenon of stormwater flooding: Case study, Environ. Model. Softw., 150, 105335, https://doi.org/10.1016/j.envsoft.2022.105335, 2022

It is OK to replicate a method, particularly if it is a part of a system (of several modules) that requires to be run to test newly proposed hypotheses and questions or in quite different catchments for the model adaptation, given both clear objectives and well-explained results. However, this manuscript lacks clear presentations of objectives, newly focused methods and results, and solid evidence of impact: e.g.,

- Some presentations of modules 1 to 8 and corresponding results were adopted too much from the two articles above with slight changes in sample events, sub-catchments, and letters in the equation without providing clear explanation written in this manuscript; e.g., specific flood volume is defined in this manuscript as $\kappa$ in eq.1 without referring as "specific flood volume", then later appears in line 97 and in line 87 as $\lambda$, (which was used and better explained in Szeląg et al. 2021). This example can be a trivial, but such way of presenting the adopted methods and results on modules 1 to 8 (Section 2 – missing explanations on DC, S1, boundary of sub-catchments, and why divided in this way; more can be found in Sections 3.1 to 3.7 as well as Sections 4.1-4.5) made the manuscript unclear and confusing if the results were obtained from this work or speculated from the previous work. This made the Section of conclusion weak as well; e.g., the authors conclude "no other previous study has included such a broad scope of analysis" (line 550), however they adopted their previous work and presented similar results here providing similar messages and interpretation.

- Module 9 (section 3 methodology, section 3.8 and section 4.7) looks newly incremented in this presented work. Although the authors mention briefly in the introduction of Methodology (line 145) its needs, this section is short and lacks clear explanation of the method and the results. In particular, section 4.7 needs better writing.
- Title and abstract address the risk assessment. However, there are no corresponding works/materials presented. Either adding more work on this part or revising the title and abstract is expected.
- It was not clearly explained on certain threshold/coefficient values mentioned in the text (e.g., the threshold value of specific flood volume used 13 $m^3 \cdot ha^{-1}$ (line 69), is this derived also from one of the modules? If the method is applied to another catchment, how this threshold should be set?
- The analyses regarding different sub-catchments (mostly Section 4.6, with the rainfall duration time of 30 min) need better explanation and writing. When selecting the different sub-catchments to decide the modernization of stormwater network in practice, would the presented set-up of comparisons in different sizes but inclusive way (e.g., wouldn't J affect M as well?) be necessary and useful?
- Too many supplement materials are added. Some may better fit to the main text (e.g., sub-catchment domain with clearer indication of boundary as used in the presented analyses).

Therefore, I would recommend the rejection of the current manuscript and encourage the authors to resubmit the improved manuscript focused on more new findings (e.g., module 9 with better explanations and interpretation of the results) and clarify module outputs from 1 to 8 more relevant to the focused work.

---

## Community Comment (CC1)

Ref: HESS_2022_109_review (submitted on 25 April 2022) Title: The role of catchment characteristics, sewer network, SWMM model parameters in urban catchment management based on stormwater flooding: modelling, sensitivity analysis, risk assessment

Dear Reviewer, Thank you for your attention. Indeed, aspects of uncertainty analysis (its effect on simulation results of specific flood volume and manhole overflow) and the use of a logistic regression model were addressed in the publications mentioned above. However, the data and context of their use was different than in the manuscript submitted for review. In connection with this, let us clarify it. Namely, in the manuscript: "Influence of urban catchment characteristics and rainfall origins on the phenomenon of stormwater flooding: A case study" (Environmental Modelling and Software) developed a logit model based on rainfall data and catchment characteristics on identifying only the phenomenon of stormwater flooding in the catchment. These analyses were performed for sub-catchments A, B, C, D (Fig. 1), which was justified by their location and the possibility of conducting a field survey simultaneously in two catchments during a single rainfall event.

[Figure]

Fig. 1. Study catchments for the determination of a logit model for stormwwater flooding identification (Szeląg et al. 2022)

The 159 rainfall events observed during the period 2008 - 2019 were used for this purpose. Although the feasibility of using the developed model to identify stormwater flooding in a catchment during a single rainfall - runoff event was demonstrated, because the field study covered only 4 catchments, the resulting relationships in the manuscript by Szeląg et al. (2022) did not fully reflect the phenomenon of stormwater flooding occurrence and did not take into account the volume of stormwater flooding. The limitations of the developed model are highlighted in this work. In the context of literature data (Jato - Espino et al. 2018, Li and Willems 2020), improved predictive capability was possible by using a hydrodynamic model which enables to perform simulations. Nevertheless, the authors focused on developing simulators to identify flooding from a single manhole based on simulation results with a calibrated hydrodynamic model. However, in the context of uncertainty analysis, it is known that there is an interaction between the calibrated parameters, resulting in many possible combinations of SWMM parameters for which identical matches between computational results and measurements are obtained. In this context, it is difficult to unambiguously consider as correct the results of a simulation of a well stormwater flooding, for example, ignoring the uncertainty of the calibrated parameters.

Once we identified this problem, we concluded that it was necessary to determine the effect of uncertainty in the calibrated SWMM model parameters on the results of the parameter calculations that form the basis for evaluating the performance of the drainage network. A literature study (Siekman and Pinekamp 2011) indicated that the appropriate parameters for evaluating the performance of a drainage system are specific flood volume and manhole overflow (degree of flooding). These calculations were made for the entire urban catchment, without sub-catchment division in the paper: "Relationship between rainfall duration and sewer system performance measures within the context of uncertainty" (Water Resources Management). However, is the value of specific flood volume determined for the whole catchment an adequate measure of stormwater system performance evaluation and should it serve as a basis for decision making? Of course not, because it is important to answer the question which part of the drainage system needs to be upgraded. To meet this objective, there would be a need to separate sub-catchments in the model where specific flood volume would be determined. In the manuscript: "Relationship between rainfall duration and sewer system performance measures within the context of uncertainty" such analyses were not performed. Indeed, in the manuscript by Szeląg et al. (2021) the influence of SWMM model parameters on specific flood volume was analyzed, but the data were presented for a single rainfall only, and thus the obtained relationships were of a preliminary nature and only illustrated certain trends, but not strict relationships that can be practically used. To determine the relationship between specific flood volume and SWMM model parameters in the work of Szeląg et al. (2021), the results of sewer operation simulations were used in which 5000 simulations were performed (considering the combination of SWMM parameters) for each independent rainfall event. The obtained preliminary results turned out to be interesting and bearing in mind that so far no simulator of specific flood volume has been developed that would simultaneously take into account the characteristics of the catchment, the stormwater network and the parameters of the SWMM model, an attempt was made to build one. For this purpose, sub-basins were separated in the model using the developed hydrodynamic model, which is a common practice. In this approach, the authors were guided by the limitations of the obtained logit model obtained in the paper: "Influence of urban catchment characteristics and rainfall origins on the phenomenon of stormwater flooding: A case study". The aim was not to simulate similar processes, because the probability of a specific flood volume and the probability of a stormwater flooding are quite different independent parameters. The use of measured data of stormwater flooding in the catchment (Szeląg 2022), as well as the adopted calculation methodology clearly confirmed the lack in the developed model.

The number and diversity of catchment characteristics used to build the model were insufficient from a simulation point of view. It is the number of data for model building that determines the results of calculations and the relationships obtained. In this context and bearing in mind that the aim of the analyses is an attempt to develop a universal tool for the use of hydrodynamic modeling, increasing the number of sub-basins for the planned experiment made it possible to increase the data for model building. We wanted to mention that the division of catchments and separation of sub-catchments was supported by the analyses of Walek (2019), who separated sub-catchments of side channels as part of his PhD dissertation and spatial data analyses for the whole of Kielce. The number of sub-catchments, their arrangement, was conditioned by the variation within them of the characteristics of the catchment, stormwater network, which is important from the point of view of the scope of applicability of the simulation model built. Taking into account the limitations of the models developed so far and the simulation results in the works of Szeląg et al. (2021, 2022), Jato - Espino et al. (2018), Li and Willems (2020), a simulation experiment was planned involving the separation in the analyzed catchment, sub-catchments for which simulations of specific flood volume were performed for the separated rainfall events in the observational series of measurements (2008 - 2018). Thus, the methodology proposed in this paper is a compilation of experiments, but not of computational results obtained in the works: ,,Influence of urban catchment characteristics and rainfall origins on the phenomenon of stormwater flooding: Case study", "Relationship between rainfall duration and sewer system performance measures within the context of uncertainty". Efforts were made to develop a model that would reflect the operating conditions of the stormwater system (in the

context of hydrology), but also to increase the amount of data to build the model, which is an indirect method to increase its accuracy (if the results of calculations with the logit model proved to be unsatisfactory, more advanced methods of machine learning would be applied). Using such a modified methodological approach to preparing data for model building, a logit model was developed that has nothing in common with the model obtained in the work of Szeląg et al. (2022).

Dear Reviewer, The objective of the present analyses was primarily to develop a tool to determine the influence and interaction between the calibrated parameters of the SWMM model and the specific flood volume taking into account both catchment characteristics, stormwater network and rainfall data. Based on the developed model, it was determined that at the stage of sensitivity analysis, boundary conditions are important. The values of catchment characteristics determine the influence of SWMM model parameters on stormwater flooding, which is very important from the point of view of model calibration, selection of techniques for identifying catchment and stormwater network characteristics before attempts are made to create a hydrodynamic model. The methodology obtained in this study actually answers a number of questions that can be initially answered before the construction of the hydrodynamic model is started. This is extremely important as it allows for optimization of the model calibration methodology. The obtained results and the model can be used as a tool for preliminary identification of requirements that must be fulfilled for the model to provide high agreement between the calculation results and measurements.

The manuscripts of Szeląg et al. (2021, 2022) only attempted a step-by-step methodology to identify the problem due to the enormous computational effort involved. As mentioned above 108 precipitation events were used to build the logit model (Szeląg et al. 2021), 90,000 simulations (16 rainfall events and 5000 simulations) were performed to determine the effect of uncertainty in the calibrated SWMM parameters, in the present problem the number of simulations is many times larger. Undertaking such a complex problem indeed required preliminary analyses, which were necessary because in undertaking subsequent simulation problems we were unable to answer the question of whether it is possible to develop such a simulator.

In this manuscript, we tried to highlight the influence of catchment characteristics on the results of sensitivity coefficient calculations, which is important from the point of view of selecting SWMM model parameters for calibration, but also may be relevant at the stage of planning the location of measuring devices. In addition, efforts were made to focus on reflecting stormwater flooding conditions in catchments given the varying catchment characteristics. From this point of view, it would indeed be reasonable to attempt to compare the results obtained with the studies of other authors and to perform a preliminary verification of the model developed. It seems advisable to compare the results of the sensitivity analysis obtained by other authors. First of all, it is advisable to highlight the influence and interaction between catchment characteristics and SWMM parameters in the context of literature data and to demonstrate the usefulness of the developed model.

We would also like to mention that the results of calculations, which were obtained in the previous works: ,,Influence of urban catchment characteristics and rainfall origins on the phenomenon of stormwater flooding: Case study", "Relationship between rainfall duration and sewer system performance measures within the context of uncertainty" were not used for model building in this manuscript. Data for building the logit model (in this manuscript) were obtained by performing independent computer simulations over a period of 4 months.

Literature

Jato-Espino, D., Sillanpää, N., Andrés-Doménech, I., Rodriguez-Hernandez, J., 2018. Flood Risk Assessment in Urban Catchments Using Multiple Regression Analysis. J. Water Resour. Plan. Manag. 144, 04017085. https://doi.org/10.1061/(ASCE)WR.1943-5452.0000874

Li, X., Willems, P., 2020. A Hybrid Model for Fast and Probabilistic Urban Pluvial Flood Prediction. Water Resour. Res. 56. https://doi.org/10.1029/2019WR025128

Siekmann, M., Pinnekamp, J., 2011. Indicator based strategy to adapt urban drainage systems in regard to the consequences caused by climate change. 12[th] International Conference on Urban Drainage, Porto Alegre (Brazil)

Szeląg, B., Kiczko, A., Łagód, G., De Paola, F. 2021. Relationship between rainfall duration and sewer system performance measures within the context of uncertainty, Water Res Manage., 35, 5073 – 5087, https://doi.org/10.1007/s11269-021-02998- x.

Szeląg, B., Suligowski, R., De Paola, F., Siwicki, P., Majerek, D., Łagód, G. 2022. Influence of urban catchment characteristics and 760 rainfall origins on the phenomenon of stormwater flooding: Case study, Environ. Model. Softw., 150, 105335, https://doi.org/10.1016/ j.envsoft.2022.105335.

Wałek, G., 2019. Wpływ dróg na kształtowanie spływu powierzchniowego w obszarze zubranizowanym na przykładzie zlewni rzeki Silnicy w Kielcach. Jan Kochanowski University Press, Kielce (in Polish)

We are also grateful for your specific remarks. We would like to address them in the author response letter.

---

## Author Comment (AC1)

This manuscript presents a model to simulate specific flood volume considering both catchment and stormwater network characteristics, and including a module to calculate and indicate possible failure of the stormwater network, which is an interesting topic. The presented model algorithms consist of nine modules (Section 3 methodology): eight modules are a replication of what were developed and published by the same lead author; i.e., Szeląg et al. 2021 (e.g., hydrodynamic model- module 2 in the manuscript, sensitivity test considering uncertainty- modules 3, 4, 6, 7) and Szeląg et al. 2022 (Logistic regression and its application to stormwater network - modules 5, 8) over the same catchment. Module 1 in this manuscript addresses 9 sub-catchments (Table 1, though mentioned as 8 in line 128) to be simulated, which the items of characteristics are applied the same way as above-mentioned articles but the resulted values shown in Table 1 are slightly different depending on the differently selected sub-catchment.

Szeląg, B., Kiczko, A., Łagód, G., De Paola, F.: Relationship between rainfall duration and sewer system performance measures within the context of uncertainty, Water Res Manage., 35, 5073 – 5087, https://doi.org/10.1007/s11269-021-02998- x, 2021

Szeląg, B., Suligowski, R., De Paola, F., Siwicki, P., Majerek, D., Łagód, G.: Influence of urban catchment characteristics and 760 rainfall origins on the phenomenon of stormwater flooding: Case study, Environ. Model. Softw., 150, 105335, https://doi.org/10.1016/j.envsoft.2022.105335, 2022

It is OK to replicate a method, particularly if it is a part of a system (of several modules) that requires to be run to test newly proposed hypotheses and questions or in quite different catchments for the model adaptation, given both clear objectives and well explained results. However, this manuscript lacks clear presentations of objectives, newly focused methods and results, and solid evidence of impact: e.g., • Some presentations of modules 1 to 8 and corresponding results were adopted too much from the two articles above with slight changes in sample events, subcatchments, and letters in the equation without providing clear explanation written in this manuscript; e.g., specific flood volume is defined in this manuscript as in eq.1 without referring as "specific flood volume", then later appears in line 97 and in line 87 as  $\lambda$ , (which was used and better explained in Szelgg et al. 2021). This example can be a trivial, but such way of presenting the adopted methods and results on modules 1 to 8 (Section 2 – missing explanations on DC, S1, boundary of sub-catchments, and why divided in this way; more can be found in Sections 3.1 to 3.7 as well as Sections 4.1-4.5) made the manuscript unclear and confusing if the results were obtained from this work or speculated from the previous work. This made the Section of conclusion weak as well; e.g., the authors conclude "no other previous study has included such a broad scope of analysis" (line 550), however they adopted their previous work and presented similar results here providing similar messages and interpretation.

**Response 1**

Dear Reviewer, Thank you for your comment. Indeed, aspects of uncertainty analysis (its effect on simulation results of specific flood volume and manhole overflow) and the use of a logistic regression model were addressed in the publications mentioned above. However, the data and context of their use was different than in the manuscript submitted for review.

In connection with this, let us clarify it. Namely, in the manuscript: "Influence of urban catchment characteristics and rainfall origins on the phenomenon of stormwater flooding: A case study" developed a logit model based on rainfall data and catchment characteristics on identifying only the phenomenon of stormwater flooding in the catchment. These analyses were performed for sub-catchments A, B, C, D (Fig. 1), which was justified by their location and the possibility of conducting a field survey simultaneously in two catchments during a single rainfall event.

Fig. 1. Study catchments for the determination of a logit model for stormwater flooding identification (Szeląg et al. 2022)

The 159 rainfall events observed during the period 2008 - 2019 were used for this purpose. Although the feasibility of using the developed model to identify stormwater flooding in a catchment during a single rainfall - runoff event was demonstrated, because the field study covered only 4 catchments, the resulting relationships in the manuscript by Szeląg et al. (2022) did not fully reflect the phenomenon of stormwater flooding occurrence and did not take into account the volume of stormwater flooding

The limitations of the developed model are highlighted in this work. In the context of literature data (Jato - Espino et al. 2018, Li and Willems 2020), improved predictive capability was possible by using a hydrodynamic model which enables to perform simulations. Nevertheless, the authors focused on developing simulators to identify flooding from a single manhole based on simulation results with a calibrated hydrodynamic model. However, in the context of uncertainty analysis, it is known that there is an interaction between the calibrated parameters, resulting in many possible combinations of SWMM parameters for which identical matches between computational results and measurements are obtained. In this context, it is difficult to unambiguously consider as correct the results of a simulation of a well stormwater flooding, for example, ignoring the uncertainty of the calibrated parameters.

Once we identified this problem, we concluded that it was necessary to determine the effect of uncertainty in the calibrated SWMM model parameters on the results of the parameter calculations that form the basis for evaluating the performance of the drainage network. A literature study (Siekman and Pinekamp 2011) indicated that the appropriate parameters for evaluating the performance of a drainage system are specific flood volume and manhole overflow (degree of flooding). These calculations were made for the entire urban catchment, without sub-catchment division in the paper: "Relationship between rainfall duration and sewer system performance measures within the context of uncertainty". However, is the value of specific flood volume determined for the whole catchment an adequate measure of stormwater system performance evaluation and should it serve as a basis for decision making?

Of course not, because it is important to answer the question which part of the drainage system needs to be upgraded. To meet this objective, there would be a need to separate subcatchments in the model where specific flood volume would be determined. In the manuscript: , "Relationship between rainfall duration and sewer system performance measures within the context of uncertainty" such analyses were not performed. Indeed, in the manuscript by Szeląg et al. (2021) the influence of SWMM model parameters on specific flood volume was analyzed, but the data were presented for a single rainfall only, and thus the obtained relationships were of a preliminary nature and only illustrated certain trends, but not strict relationships that can be practically used. To determine the relationship between specific flood volume and SWMM model parameters in the work of Szeląg et al. (2021), the results of sewer operation simulations were used in which 5000 simulations were performed (considering the combination of SWMM parameters) for each independent rainfall event.

The obtained preliminary results turned out to be interesting and bearing in mind that so far no simulator of specific flood volume has been developed that would simultaneously take into account the characteristics of the catchment, the stormwater network and the parameters of the SWMM model, an attempt was made to build one. For this purpose, sub-basins were separated in the model using the developed hydrodynamic model, which is a common practice. In this approach, the authors were guided by the limitations of the obtained logit model obtained in the paper: , "Influence of urban catchment characteristics and rainfall origins on the phenomenon of stormwater flooding: A case study". The aim was not to simulate similar processes, because the probability of a specific flood volume and the probability of a stormwater flooding are quite different independent parameters. The use of measured data of stormwater flooding in the catchment (Szeląg 2022), as well as the adopted calculation methodology clearly confirmed the lack in the developed model.

Taking into account the limitations of the models developed so far and the simulation results in the works of Szeląg et al. (2021, 2022), Jato - Espino et al. (2018), Li and Willems (2019), a simulation experiment was planned involving the separation in the analyzed catchment, sub-catchments for which simulations of specific flood volume were performed for the separated rainfall events in the observational series of measurements (2008 - 2018). Thus, the methodology proposed in this paper is a compilation of experiments, but not of computational results obtained in the works: "Influence of urban catchment characteristics and rainfall origins on the phenomenon of stormwater flooding: Case study", "Relationship between rainfall duration and sewer system performance measures within the context of uncertainty". Efforts were made to develop a model that would reflect the operating conditions of the stormwater system (in the context of hydrology), but also to increase the amount of data to build the model, which is an indirect method to increase its accuracy (if the results of calculations with the logit model proved to be unsatisfactory, more advanced methods of machine learning would be applied). Using such a modified methodological approach to preparing data for model building, a logit model was developed that has nothing in common with the model obtained in the work of Szeląg et al. (2022).

Dear Reviewer, The objective of the present analyses was primarily to develop a tool to determine the influence and interaction between the calibrated parameters of the SWMM model and the specific flood volume taking into account both catchment characteristics, stormwater network and rainfall data. Based on the developed model, it was determined that at the stage of sensitivity analysis, boundary conditions are important. The values of catchment characteristics determine the influence of SWMM model parameters on stormwater flooding, which is very important from the point of view of model calibration, selection of techniques for identifying catchment and stormwater network characteristics before attempts are made to create a hydrodynamic model. The methodology obtained in this study actually answers a number of questions that can be initially answered before the construction of the hydrodynamic model is started. This is extremely important as it allows for optimization of the model calibration methodology. The obtained results and the model can be used as a tool for preliminary identification results and measurements.

The manuscripts of Szeląg et al. (2021, 2022) only attempted a step-by-step methodology to identify the problem due to the enormous computational effort involved. As mentioned above 108 precipitation events were used to build the logit model (Szeląg et al. 2021), 90,000 simulations (16 rainfall events and 5000 simulations) were performed to determine the effect of uncertainty in the calibrated SWMM parameters, in the present problem the number of simulations is many times larger.

Undertaking such a complex problem indeed required preliminary analyses, which were necessary because in undertaking subsequent simulation problems we were unable to answer the question of whether it is possible to develop such a simulator.

In this manuscript, we tried to highlight the influence of catchment characteristics on the results of sensitivity coefficient calculations, which is important from the point of view of selecting SWMM model parameters for calibration, but also may be relevant at the stage of planning the location of measuring devices. In addition, efforts were made to focus on reflecting stormwater flooding conditions in catchments given the varying catchment characteristics. From this point of view, it would indeed be reasonable to attempt to compare the results obtained with the studies of other authors and to perform a preliminary verification of the model developed. It seems advisable to compare the results of the sensitivity analysis obtained by other authors. First of all, it is advisable to highlight the influence and interaction between catchment characteristics and SWMM parameters in the context of literature data and to demonstrate the usefulness of the developed model.

We would also like to mention that the results of calculations, which were obtained in the previous works: "Influence of urban catchment characteristics and rainfall origins on the phenomenon of stormwater flooding: Case study", "Relationship between rainfall duration and sewer system performance measures within the context of uncertainty" were not used for model building in this manuscript. Data for building the logit model (in this manuscript) were obtained by performing independent computer simulations over a period of 4 months.

"Climate change and urbanisation are major drivers of increased frequency and severity of hydraulic overloads in urban catchments, leading to flooding events, which cause decrease of life standard, material losses, traffic difficulties etc. (Petit-Boix et al., 2017; Chang et al., 2021). Therefore, criteria for assessing stormwater network operating were introduced, which should be taken into consideration both at the design stage and while planning corrective actions. According to these criteria, one of the key assessment parameters is a maximum number of stormwater flooding in return period (DWA – A118E, 2006; EN – 752, 2006). However, since this parameter has a typically qualitative character, some further modifications were proposed. Based on computation results for stormwater networks, Siekmann and Pinekamp (2011) defined the boundary values of specific flood volume which expressed the volume of stormwater per unit impervious area. Exceeding these values should be considered as a clear signal for decision-makers to implement the process of improving stormwater management in the catchment.

Mathematical modelling of the stormwater network provides significant support in a decisionmaking process. According to (Kirshen et al., 2015; Chen et al., 2016; Mignot et al., 2019), hydrodynamic catchment models are usually applied. For many years, the United States Environment Protection Agency (EPA) has been developing computation tools to simulate stormwater network operation. One of the most common tools is the SWMM (Storm Water Management Model) program, (Baek et al., 2020; Behrouz et al., 2020). SWMM can be applied for a simplified simulation of stormwater runoff from a catchment, hydrographs in stormwater network as well as simulation of hydraulic overloads resulting in flooding (Teng et al., 2017; Cheng et al., 2019). Simulation results (hydrographs, specific flood volumes) can be burdened with uncertainty due to lack of data on catchment spatial development, stormwater network characteristics as well as due to limited number of rainfall – runoff episodes (high resolution measurements of rainfalls and flow rates), and interaction between identified parameters.

In order to reduce uncertainty, optimization of calibrated model parameters is applied (De Paola et al., 2016; Swathi et al., 2018; Awol et al., 2018). Currently, integrating sensitivity and uncertainty analysis, which is characteristic for mechanistic models, gains the highest popularity. In case of machine learning methods (Ke et al., 2020) identification of model structure requires implementation of advanced optimization algorithms (Mignot et al., 2019), while for sensitivity analysis Cateri Paribus, Shapley index (Yang et al., 2020) methods are used. According to literature reports (Cristiano et al., 2019) in most studies rainfall intensity impact is neglected at sensitivity analysis stage. This assumption leads to generalization. Moreover, it is contrary to the findings of Fraga et al. (2016) and Cristiano et al. (2016) who proved the significance of rainfall distribution to the following relationships: catchment characteristics – peak flow, specific flood volume – calibrated parameters of SWMM model etc. Simulations of outflow hydrographs, stormwater flooding with mechanistic models for urban catchments with diverse characteristics (different surface, imperviousness, density of stormwater network etc) showed huge discrepancies. Identification of relationship between SWMM and stormwater flooding is of high importance in the context of interactions between their numerical values, which was proved in a number of previous studies (Huang et al., 2018; Xingh et al., 2021). According to literature (Dotto et al. 2014; Teweldebrhan et al., 2020; Chen et al. 2018), the most commonly used method of uncertainty analysis in urban catchments is GLUE (Generalized Likelihood Uncertainty Estimation) method. Szeląg et al. (2021) confirmed the impact of SWMM parameters uncertainty on computation of specific flood volume and of manholes overflow (without division into sub-catchments).

Due to the aforementioned circumstances, machine learning methods were applied to simulation of stormwater overflow. Thorndahl et al. (2008) based on simulation results of stormwater overflow from manholes, including uncertainty of calibrated parameters elaborated a model using FORM method. Jato-Espino et al. (2018) and Li and Willems (2020), conducting simulations with mechanistic models, appointed models for identification of overflow from a single manhole based on rainfall frequency, catchment characteristics and stormwater network characteristics. Application of those models for stormwater network management was limited and required huge burden of work as well as highly relevant data. Therefore, Szelag et al. (2022) elaborated a model for identification of stormwater flooding in a catchment, however due a number of data used during model construction, the obtained relationship had a limited application. In the aforementioned models, interactions between catchment and stormwater network characteristics and catchment retention and stormwater system conductivity were neglected. In the context of catchment management, these factors are relevant for selection of the optimal solution (green infrastructure, channel retention) of amelioration of stormwater system operation. Neglecting of a single of these factors results in a reduced applicability of the obtained tool at the stage of spatial planning. Mechanistic models include the listed above conditions, however they require detailed data and can be used only for a specific catchment. Simulator is lacking, which would at the same time include catchment characteristics, stormwater network characteristics as well as identified parameters of mechanistic model, and can be applied for different urban catchments without the need of calibration.

[revised manuscript text omitted]

In our study, a novel algorithm for creating a simulator to predict specific flood volume is developed. In the proposed approach, the stormwater flooding is related to rainfall data, catchment characteristics and calibrated SWMM model parameters, which enables this tool to be implemented for different catchments with various characteristics, both at the stage of spatial planning, corrective actions of stormwater network (optimization of canal retention and terrain retention) and during the daily operation of stormwater networks. Therefore, the proposed model is an alternative to SWMM model, does not require calibration and allows for assessment of stormwater system operation even in case of limited data set. In the adopted algorithm, an innovative sensitivity coefficient was defined, allowing for analysis of dependencies between SWMM parameters (width of runoff path, retention of impervious areas, Manning roughness coefficient of canals) on specific flood volume for adopted rainfall data and catchment characteristics. Procedures in the algorithm allow for analysis of stormwater network operation stages of the algorithm based on the measurement data from an urban catchment in Kielce are presented in the article."

**Some presentations of modules 1 to 8 and corresponding results were adopted too much from the two articles above with slight changes in sample events**

**Response 2**

Dear Reviewer, it is a fact the algorithms are similar, but they involve different data. The data for building the logistic regression model to identify the stormwater flooding (Szeląg et al. 2022) is different from that for simulating the specific flood volume. It is clear from the comments cited above that the calculation results reported in Szeląg et al. (2021, 2022) are different than those presented in this manuscript. Nevertheless, to make this clear and readable in the revised manuscript it is planned to highlight the previous calculation results and refer to the algorithms developed in the previous manuscripts. Proposals for the planned revisions are given below.

"An innovative algorithm for creating a simulator to identify a specific flood volume was proposed (Fig. 2). In contrast to existing algorithms in use, in the adopted approach, a wider scope of computation was applied, linked with analysis of model sensitivity and the impact of uncertainty of calibrated SWMM model parameters on the probability of a stormwater network failure

---

## Author Comment (AC3)

**Review 1**

*Title: The title is long, unclear and very descriptive. Furthermore, it does not reflect the main contribution of this research, which I believe is: "In our study, a novel algorithm for creating a simulator to predict specific flood volume is developed."*

**Response 1**

Dear Reviewer, Thank you for your comment. Actually, the purpose of the paper was to develop an advanced simulator of specific flood volume and to determine the influence of catchment characteristics and rainfall data on the results of the sensitivity analysis. With this fact in mind, we plan to modify the current title to read: , *"An advanced algorithm for computing stormwater flooding volume: simulation, sensitivity analysis"*. We hope that the new title fully reflects the content of our study and at the same time is it is more clear.

**Review 2**

*Line 42: »One of the main factors leading to hydraulic overloads is associated with the wearing of storm sewers resulting in increased roughness.« Please provide a reference for this statement*

**Response 2**

Dear Reviewer, Thank you for your comment. The increase in channel wall roughness is a result of both abrasion and sediment deposition. The effect of uncertainty in hydraulic conditions in channels, the amount of sediment deposited in them, on the increase in channel wall roughness was analyzed in detail by Shirazi et. al. (2014), Caradot et al. (2017). In addition, Bijnen et al. (2012) performing hydraulic analyses of sewer network performance showed the influence of hydraulic conditions in the sewers (increase in roughness due to abrasion, deposited sediments) on the results of sewer spill simulations.

**Review 3**

*Line 46: You comment that the frequency of stormwater flooding has a typically qualitative character. Can you please explain this? In my opinion, it has a quantitative character, as can be found in the mentioned standards.*

**Response 3**

Dear Reviewer, Thank you for your comment. When we used the term quality variable, we had in mind the phenomenon of stormwater flooding itself. Stormwater flooding may or may not occur. In the context of catchment management, sewage network corrective actions (implementation of green infrastructure, construction of reservoirs, etc.), the extent of flooding (area, depth, volume) and its impact on the living conditions of society and the environment are important. The planned modifications include a revision of the manuscript (text below).

,,Climate change and urbanisation are major drivers of increased frequency and severity of hydraulic overloads in urban catchments, leading to flooding events, which cause decrease of life standard, material losses, traffic difficulties etc. (Petit-Boix et al., 2017; Chang et al., 2021). One of the main factors leading to hydraulic overloads is associated with the wearing of storm sewers resulting in increased roughness (Bijnen et al., 2012; Caradot et al., 2017). Therefore, criteria for assessing

stormwater network operating were introduced, which should be taken into consideration both at the design stage and while planning corrective actions. According to these criteria, one of the key assessment parameters is a maximum number of stormwater flooding in return period (DWA – A118E, 2006; EN – 752, 2006). Although the above criterion is quantitative in nature, it does not express volume, area, depth of the stormwater flooding (…), what is important plays a role in the decision to corrective actions the sewage network. Based on computation results for stormwater networks, Siekmann and Pinekamp (2011) defined the boundary values of specific flood volume which expressed the volume of stormwater flooding per unit impervious area."

**Review 4**

*Line 65: You state that model calibration consists of two stages: sensitivity analysis and uncertainty. This might be true in mechanistic modelling. However, this is not the case for data-driven methods, which can be also used for model calibration. Please refine this statement*

**Response 4**

Dear Reviewer, Thank you for your comment. Of course sensitivity and uncertainty analysis is implemented for mechanistic models. For models based on machine learning methods other simulation approaches are implemented such as Cateri Paribus analysis, GSA, Shapley index. A modification of the manuscript is planned (text below)

,,In case of machine learning methods (Ke et al., 2020) identification of model structure requires implementation of advanced optimization algorithms (Mignot et al., 2019), while for sensitivity analysis Cateri Paribus (Wang et al. 2020), Shapley index (Yang et al., 2020), GSA (Global Sensitivity Analysis; Saltelli et al. 2000) methods are used"

Saltelli, S. Tarantola, F. Campolongo. (2000). Sensitivity analysis as an ingredient of modelling. Statistical Science, 15(4), 377–395.

Wang, D., Sven Thunéll, S., Lindberg, U., Jiang, L., Trygg, J., Tysklind, M., Souihi, N. (2020). A machine learning framework to improve effluent quality control in wastewater treatment plants. Science of the Total Environment, 784, 147138.

Yang, Y., Chui, T.F.M.: Modeling and interpreting hydrological responses of sustainable urban drainage systems with explainable machine learning methods, Hydrol. Earth Syst. Sci., 25, 5839–5858, https://doi.org/10.5194/hess-25-5839-2021, 2020.

**Review 5**

*Line 70: »Sensitivity analysis is limited to sub-catchments, and so it is impossible to predict the impact of catchment characteristics on calculation results.« This sentence is unclear. Since the catchment consists of sub-catchment they should reflect the characteristics of the catchment. Please clarify.*

**Response 5**

Dear Reviewer, Thank you for your comment. Analyzing the results of the sensitivity analysis calculations for different urban catchments (Fraga et al. 2016; Freni et al., 2012) one can find, a large variation in the simulation results. In some catchments, the greatest influence on catchment outflow/flooding volume is the roughness coefficient of impervious areas, and in other cases the Manning roughness coefficient of channel. Therefore, it seemed expedient to develop a methodology that would allow for the adopted values of catchment characteristics, sewer network to determine the influence of calibrated parameters on the results of simulation of stormwater flooding volume. Currently, sensitivity analysis, if performed at the stage of hydrodynamic model calibration, calculations are usually performed for the cross-section closing the catchment. The literature review shows (Fatonet al. 2021; Gupta and Razhavi 2016; Cristiano et al., 2018) that in most of the studies the sensitivity variability (influence of the calibrated parameters on the flooding volume) at the catchment scale is neglected. Therefore, to make this possible, sub-catchments characterized by different land use, sewerage network were separated in the considered catchment.

**Review 6**

*Line 84: "The calibration of such a model is simpler in comparison to hydrodynamic models due to the fact that a number of advanced statistical methods are already implemented in computing packages.« Aren't all of these hydrodynamic models? Please clarify*

**Response 6**

Dear Reviewer, Thank you for your comment. Of course, hydrodynamic models are mechanistic models and models based on machine learning methods. The difference is that mechanistic models and models based on machine learning methods are calibrated differently.

**Review 7**

*Line 145: "A failure was defined as exceedance of certain specific flood volume which points out that modernisation of the stormwater network is necessary." Maybe replacing old pipes with bigger ones is note the only and the most efficient solution. Perhaps a different rainwater management approach should be used (e.g., SUDS). Please rephrase accordingly.*

**Response 7**

Dear Reviewer, Thank you for your comment. Modified the term failure to read: A failure is defined as a state of operation of a stormwater drainage system (assumed rainfall load) in which hydraulic overloading occurs, channel capacity is exceeded, resulting in a specific flood volume of not less than 13 $m^3 \cdot ha^{-1}$. This requires corrective actions of the system and reduction of the runoff from

the catchment by implementing rainfall management systems (alternatively improving the efficiency of the existing type of permeable surfaces, rainwater reservoirs, etc.), increasing sewer retention. We agree, that the term "modernization" is a oversimplification and in the revision a more general statement should be used, e.g. "indicates that corrective actions have been taken, such as using larger diameter pipes or increasing water retention in the upper catchment area".

**Review 8**

*Line 146: I apologize if I missed this information somewhere earlier. At this point, we do not know yet what a "unit" is/represents. Please add an explanation.*

**Responses 8**

Dear Reviewer, Thank you for your comment. Indeed, definitions like unit volume may have appeared in the manuscript, but it is the same as specific flood volume. This will be corrected at the manuscript revision stage.

**Review 9**

*Line 151: "The proposed computation algorithm consists of 11 modules.« It is unclear, which are the 11 modules. In Figure 2 we only see 9 modules. Section 3 also ends with Subsection 3.8. (Module 9). Please clarify.*

**Responses 9**

Dear Reviewer, Thank you for your comment. Of course, the developed algorithm includes 9 computational modules.

**Review 10**

*Line 169: Please add an explanation of what is "zero-one".*

**Responses 10**

Dear Reviewer, Thank you for your comment. The logistic regression method used in this paper belongs to classification models. In this approach, the dependent variables underlying its development include 1 (when the SWMM-calculated value of the specific flood volume exceeds $13 m^3 \cdot ha^{-1}$) or 0 (the SWMM-calculated value of the specific flood volume does not exceed $13 m^3 \cdot ha^{-1}$).

**Review 11**

*Line 185: Please provide an explanation of what is C.*

**Reponses 11**

Dear Reviewer, Thank you for your comment. C - return period. This explanation will be added in the revised version of the manuscript.

**Review 12**

*Lines 180 and 188: Rainfall data for periods 2010 – 2019 and 2010 – 2018 are mentioned. Is this actually the same time period and is this just a mistake? Please clarify.*

**Responses 12**

Dear Reviewer, Thank you for your comment. Of course it is the same period (2010 - 2019). This will be clarified in the revised version of the manuscript.

**Review 13**

*Line 238: Please better explain the meaning of numbers: 200, 5000, and 9*

**Responses 13**

Dear Reviewer, Thank you for your comment. 200 is the number of separated rainfall events; 5000 is the number of simulations of a single rainfall episode taking into account uncertainty, 9 is the number of sub-basins separated in the catchment constituting the basis for the logit model determination. It will be explained in the revised version of the manuscript.

**Review 14**

*Line 319 and 313: If the terms "failure" and "breakdown" are referring to the flooding of the stormwater system, I would propose that you use the same and most clear word everywhere.*

**Responses 14**

Dear Reviewer, Thank you for your comment. We will, of course, standardize the nomenclature throughout the manuscript if possible.

**Review 15**

*Line 323: Please clarify what is MC.*

**Responses 15**

Dear Reviewer, Thank you for your comment. MC means Monte Carlo (random number generator)

**Review 16**

*Line 349: Please use the same abbreviation for the likelihood function in Figs. S3- S10, and Section 3.3.*

**Responses 16**

Dear Reviewer, Thank you for your comment. This will be included in the manuscript

**Review 17**

*Lines (371-374): It is not clear how equations 9-12 were derived. Please provide an explanation.*

**Responses 17**

Dear Reviewer, Thank you for your comment. In order to clarify the equations, the description of the logistic regression method was modified. The proposed modification is included below.

3.5. Developing a logistic regression model (module 5)

Logistic regression, also known as the logit model, is a tool used for classification. Its benefits in comparison to widely used methods arise from the fact that computation results range from 0 to 1 and they represent probability values. This model has been already applied for modelling storm overflows (Szeląg et al., 2020), identifying stormwater flooding from manholes (Jato – Espino et al., 2018) and the technical condition of sewage systems (Salman and Salem, 2012). The logistic regression model is described by the following equation:

$$p_m = \frac{\exp(\alpha_0 + \alpha_1 \cdot x_1 + \alpha_2 \cdot x_2 + \alpha_3 \cdot x_3 + \cdots + \alpha_i \cdot x_i)}{1 + \exp(\alpha_0 + \alpha_1 \cdot x_1 + \alpha_2 \cdot x_2 + \alpha_3 \cdot x_3 + \cdots + \alpha_i \cdot x_i)} = \frac{\exp(X)}{1 + \exp(X)} = \frac{exp(X_{rain} + X_{SWMM} + X_{Catchm})}{1 + exp(X_{rain} + X_{SWMM} + X_{Catchm})} \qquad (4)$$

where $p_m$ – probability of a specific flood volume (understood as the need to corrective actions the stormwater network); $\alpha_0$ – absolute term; $\alpha_1$, $\alpha_2$, $\alpha_3$, $\alpha_i$ – values of coefficients estimated with the maximum likelihood method, X – vector describing the linear combination of the independent variables; $X_{rain}$ – vector describing linear combination of t statistically significant rainfall characteristics $(X_{rain} = \sum_{s=1}^{t} \alpha_s \cdot x_s)$; $X_{SWMM}$ – vector describing linear combination of m statistically significant SWMM parameters $(X_{SWMM} = \sum_{k=1}^{m} \alpha_k \cdot x_k)$; $X_{Catchm}$ – vector describing linear combination of r statistically significant catchment characteristics, and stormwater network characteristics confidence level – 0.05 $(X_{Catchm} = \sum_{p=1}^{r} \alpha_p \cdot x_p)$; $x_i$ – independent variables describing rainfall characteristics, e.g., rainfall depth, its duration, and the parameters calibrated in the SWMM, catchment characteristics (permeability, terrain retention, density of stormwater network, length, slope, retention in stormwater channels etc.). Independent variables in the logit model were calculated using the forward stepwise algorithm, recommended for the creation of such models. At the same time, it also ensures the elimination of correlated independent variables (Harrell 2001). The estimation of the coefficients $\alpha_i$ in equation (4) and thus the determination of the logistic regression model involved two stages: learning (80% of the data i.e. 7.200.000) and testing (20% of the data i.e. 1.800.000).

4.3. Determination of the logistic regression model (module 5)

A logistic regression model was built based on the operational simulation of the stormwater network. The model can be used to identify specific flood volume and for decision-making regarding corrective actions

of the stormwater system. The relationship from Equation (3) can be described by the following linear combination:

$$X_{rain} = 4.05 \cdot P_{tot} - 0.18 \cdot t_r - 54.15 \tag{10}$$

$$X_{SWMM} = 0.23 \cdot \alpha - 79.40 \cdot n_{imp} + 6.23 \cdot \beta + 0.33 \cdot \gamma + 234.12 \cdot n_{sew} \tag{11}$$

$$X_{Catchm} = 76.72 \cdot Imp + 40.77 \cdot Impd - 0.01 \cdot Vk - 1967.04 \cdot Gk - 1169.00 \cdot Gkd - 20.33 \cdot Jkp \tag{12}$$

For other independent variables (Tab. S2) the determined coefficients appeared to be statistically insignificant in prediction confidence band 0.05. Standard deviations of the coefficients estimated from the logit model and the test probabilities are presented in Tab. S2. The performed computations indicated that the best fit of the computed results to the measurement data was obtained for $p_{m,cr}$ = 0.75. For the test data set (20%) the following values were obtained: SPEC = 95.24%, SENS = 84.62% and Acc = 87.87%.

**Review 18**

*Line 384: You refer to Fig. 3, but it seems you are referring to Fig. 4. Please check*

**Responses 18**

Dear Reviewer, Thank you for your comment. We apologize for the mistake, of course it should be figure 4, this will be corrected.

**Review 19**

Line 505 – 507: "However, to date, there has been no statistical model that would take into account both hydrodynamic model parameters as well as catchment and stormwater network characteristics.« Please clarify, aren't the parameters of the hydrodynamic models reflecting (i.e., are the same as) catchment and stormwater network characteristics?

**Responses 19**

Dear Reviewer, Thank you for your comment. In the context of the above comments, we meant that hydrodynamic models based on machine learning methods do not simultaneously take into account rainfall data, catchment characteristics, sewer network, catchment retention, retention and sewer network capacity. The inclusion in the developed simulator of catchment characteristics (impervious area), sewer network (sewer network density, retention, channel longitudinal slope), area retention (Manning's roughness coefficient of impervious areas, retention depth of impervious area, correction coefficient for percentage area) and channel capacity (Manning's roughness coefficient) makes it possible to apply the model to catchment management. Thus, the obtained simulator is an alternative to the used mechanistic models and does not require calibration, which is important from the point of view of implementation for catchments with different characteristics.

**Review 20**

*Line 548: Please clarify what are lower and upper parts.*

**Responses 20**

Dear Reviewer, Thank you for your comment. We had in mind the catchment area and the lower and upper streams

**Review 21**

*Lines 553, 554: You refer to the proposed algorithm as a tool. Please use one word consistently. As mentioned before, if would be beneficial to reconsider the article title accordingly.*

**Responses 21**

Dear Reviewer, Thank you for your comment. The vocabulary will be standardized. As suggested by the Reviewer, the title of the article has been corrected.

**Review 22**

*Discussion and Conclusions: Some of the results presented in this article are expected and not so novel for the urban drainage modelling community (e.g., impervious area leads to an increase of flooding).*

**Responses 22**

Dear Reviewer, Thank you for your comment. In accordance with the above comments, we have planned to revise the discussion and conclusions by reducing repetition. The difference between the models developed so far and the one presented in this manuscript has been highlighted. The influence of catchment characteristics and rainfall data on the obtained relationships between specific flood volume and calibrated SWMM model parameters was noted. Proposed corrections are provided below.

Developing and calibrating mathematical models to simulate stormwater network operation under hydraulic overloads is one of the latest areas of research. In comparison to the statistical models used so far (Li and Willems, 2019; Thorndahl 2009), the approach proposed in our study includes SWMM model parameters describing catchment retention and, at the same time, the characteristics of the catchment and stormwater network (tab. S4). Apart from the model developed in this study, the above mentioned factors are only included in mechanistic models, which have a form of differential equations. Therefore, they require a large number of simulations in order to determine the impact of selected variables on computation results of specific flood volume. Models developed with machine learning methods are free of such drawbacks (tab. S4), which have a form of empirical relationships. In contrast, in case of models developed with neuron networks, there is a need of performing additional analyses (Ke et al, 2020; Yang et al., 2020).

**Tab. S4. Comparison of developed model for identification of specific flood volume to literature data**

| Study | Criteria | M | I | R | C | S | P |
|---|---|---|---|---|---|---|---|
| Duncan et al. (2011) | occurence of flooding | ✓ | • | ✓ | ✓ | ✓ | • |
| Jato - Espino et al. (2018) | occurence of flooding | ✓ | ✓ | ✓ | ✓ | ✓ | • |
| Jato - Espino et al. (2019) | occurence of flooding | ✓ | • | ✓ | ✓ | ✓ | • |
| Li and Willems (2020) | occurenceof flooding | ✓ | ✓ | ✓ | ✓ | ✓ | • |
| Szeląg et al. (2022) | occurence of flooding | ✓ | ✓ | ✓ | ✓ | ✓ | ✓ |
| Szeląg et al. (2021) | volume | • | • | ✓ | ✓ | ✓ | ✓ |
| Thorndahl et al. (2008) | volume | ✓ | ✓ | ✓ | • | ✓ | ✓ |
| Verbovski et al. (2022) | volume | ✓ | ✓ | ✓ | • | • | • |
| Fu et al. (2011) | volume | • | • | ✓ | ✓ | ✓ | ✓ |
| Chen et al. (2020) | volume | • | • | ✓ | ✓ | ✓ | ✓ |
| Fraga et al. (2016) | volume | • | • | ✓ | ✓ | ✓ | ✓ |
| this study | volume | ✓ | ✓ | ✓ | ✓ | ✓ | ✓ |

where: M (method); the models were divided into two groups: mechanistic (·) and machine learning (ᵛ); R (rainfall); C (catchment); S (sewer); P (calibration parameter); I (interpretation model, based on estimated factors the impact of analysed factors on stormwater flooding can be determined).

Jato – Espino et al. (2018, 2019) and Li and Willems (2020) analysed stormwater flooding from manholes based on catchment characteristics and stormwater network characteristics (tab S4). Szeląg et al. (2022) confirmed their results and developed a model for identification of stormwater flooding in a catchment. Besides, by indicating the impact of uncertainty of SWMM model parameters on stormwater flooding, Szeląg et al. (2021) proved that previous approaches require further development . In the wider context of catchment management, their approach does not apply for the characteristic of the materials used for road, roofs or parking places, etc. Fu et al. (2011) and Thorndahl et al. (2009) analyzed the uncertainty of the identified parameters, which allowed, for example, to correct for impervious area retention, roughness coefficient without being able to correct for catchment imperviousness, which limited the use of the models in catchment management. The approach proposed in our study is a combination of these two solutions, which provides a tool which can be successfully implemented to manage other catchments.

The results of our study confirmed the major significance and huge interaction between catchment characteristics and SWMM model parameters. This fact can be further compared by several references (Li and Willems, 2020; Jato – Espino et al., 2019; Zhuo et al., 2019) presenting comparisons of flooding simulations in urban catchments. Our analysis indicated that an impervious area in a catchment (Imp, Impd) leads to the increase of flooding; reverse dependency was obtained by Jato – Espino et al. (2018) when modelling flooding from manholes. Increase in channel volume above the closing cross-section of a catchment (Vk) and its longitudinal slope (Jkp) results in the decrease of flooding, that was confirmed by computations performed for Espoo catchment in Finland (Jato – Espino et al. 2019). Interestingly, the increase of unit impervious area per the length of main stormwater interceptor (Gk, Gkd) results in smaller volume of stormwater flooding. This result is absolutely right due to the fact that the longer the channel, the greater the number of manholes.

Huang et al. (2018) based on observations conducted in a complex stormwater system indicated the impact of catchment location and hydrological conditions on the peak flow. Yao et al. (2019) obtained similar results after computations with a mechanistic model for catchments in Beijing and in Dresden (Reyes – Silva et al. 2020).

Calculation results obtained in this study confirmed relevant impact of rainfall data, catchment characteristics, and stormwater network characteristics on sensitivity coefficients – relationships between SWMM parameters and specific flood volume. For rainfall data and catchment characteristics (assumed as constant) it was proved that correction coefficient of impervious area ($\beta$) and the Manning roughness coefficient for channels ($n_{sew}$) have the greatest impact on specific flood volume. The results of our computations are consistent with Thorndahl et al. (2009), who simulate flooding from a single manhole in the Frejlev catchment (Belgium), based on rainfall data and calibrated parameters of a mechanistic model. These findings were confirmed by calculations Fu et al. (2012) and Prodanovic et al. (2022) respectively for catchments of 400 ha and 8 ha. Szeląg et al. (2021) based on simulations with mechanistic model including uncertainty of SWMM parameters proved the key impact of Manning roughness coefficient of sewers on specific flood volume (for rainfall episod $t_r$ = 30 min and $P_t$ = 15.25 mm). Fraga et al. (2016) used GLUE+ GSA method for a small road catchment and indicated the impact of rainfall data (rainfall duration, depth, temporal distribution) on sensitivity analysis results. I was further confirmed in computations of stormwater flooding using logit model (Szeląg et al. 2022) and specific flood volume calculations with SWMM model (Freni et al. 2012). Xing et al. (2021) used mechanistic model to determine characteristics of spatial development and stormwater characteristics in Chongqing catchment (China) on the depth of stormwater flooding. The aforementioned research studies indicate the impact of rainfall data, catchment characteristics, and stormwater network characteristics on sensitivity of hydrodynamic simulation model for stormwater flooding.

Differences in probability of specific flood volume/sensitivity coefficients indicate the influence of catchments downstream on conditions in the catchment above. The variation in sensitivity coefficients does not account for local conditions within the side channels. Due to the creation of successive sub-catchments by combining them, the conditions of the sewer system in its area are averaged out, making the interpretation of the results difficult. Using the developed tool, catchment management may become difficult when there is a particularly hydraulically overloaded area within the catchment, which impacts neighboring sub-catchments.